# Overcoming thermostability challenges in mRNA–lipid nanoparticle systems with piperidine-based ionizable lipids
Kazuki Hashiba [1], Masamitsu Taguchi[1], Sachiko Sakamoto[1], Ayaka Otsu[1], Yoshiki Maeda[1], Hirofumi Ebe[1], Arimichi Okazaki[1], Hideyoshi Harashima [2] & Yusuke Sato [2]

Lipid nanoparticles (LNPs) have emerged as promising platforms for efficient in vivo mRNA delivery owing to advancements in ionizable lipids. However, maintaining the thermostability of mRNA/LNP systems remains challenging. While the importance of only a small amount of lipid impurities on mRNA inactivation is clear, a fundamental solution has not yet been proposed. In this study, we investigate an approach to limit the generation of aldehyde impurities that react with mRNA nucleosides through the chemical engineering of lipids. We demonstrated that piperidine-based lipids improve the long-term storage stability of mRNA/LNPs at refrigeration temperature as a liquid formulation. High-performance liquid chromatography analysis and additional lipid synthesis revealed that amine moieties of ionizable lipids play a vital role in limiting reactive aldehyde generation, mRNA–lipid adduct formation, and loss of mRNA function during mRNA/LNP storage. These findings highlight the importance of lipid design and help enhance the shelf-life of mRNA/LNP systems.

mRNA has recently gained increasing importance and popularity because of its numerous advantages, including the ability to encode specific proteins and the potential for rapid development and production, such as the marketed mRNA-based vaccines developed against SARS-CoV-2[1,2].

Lipid nanoparticles (LNPs) are widely employed for in vivo delivery of mRNA; they protect the mRNA payload from degradation by nucleases and facilitate their delivery to target cells[3]. LNPs typically consist of various components, such as ionizable lipids, cholesterol, phospholipids, and polyethylene glycol (PEG)-conjugated lipids. Among these components, ionizable lipids contain a positively charged amino group that interacts with the negatively charged phosphate backbone of mRNA, thereby playing a critical role in encapsulating the mRNA and promoting its escape from the endosome[4–6]. The evolution of ionizable lipids has enabled efficient in vivo mRNA delivery and broadened their applications in cancer vaccines, protein replacement therapy, and genome editing[7,8].

The long-term storage of mRNA/LNP formulations remains a significant challenge[9,10]. mRNAs are inherently susceptible to degradation via in-line hydrolytic cleavage and oxidation[11,12]. In addition, LNPs and their components can undergo physical and chemical damage during storage. These degradation processes are accelerated by thermal stress[13]. To mitigate these risks, mRNA/LNP formulations are stored at −20 °C or below. The

COVID-19 pandemic led to the establishment of cryogenic storage facilities and logistics systems worldwide; however, it is preferable to avoid cryogenic temperatures for the widespread use of sustainable mRNA therapeutics owing to cost considerations.

Lyophilization is a promising approach for improving the thermostability of mRNA/LNP[13–16]. Reducing the amount of residual water in the LNP core decreases the risk of mRNA degradation[17], but it requires additional complex, lengthy, and high-energy processes. In addition, the nanoparticles can be potentially damaged and physicochemically changed by reconstitution, even in the presence of cryoprotectants.

There is a growing demand for thermostable mRNA/LNPs that can be stored in liquid form. It is important to overcome the innate instability of mRNA along with another significant hurdle, that is, an unintended addition reaction of lipid impurities with mRNA. Ionizable lipids with tertiary amines generate aldehyde impurities through oxidation and hydrolysis; these impurities covalently bind to mRNA, compromising their integrity and activity during storage[18]. This mechanism decreases the thermostability of mRNA/LNP systems. The amine head of ionizable lipids is responsible for aldehyde generation; therefore, we hypothesized that the oxidation and hydrolysis of ionizable lipids and the resulting aldehyde generation could be limited by strategically designing the amine structure; this could enhance the

[1]Nucleic Acid Medicine Business Division, Nitto Denko Corporation, 1-1-2, Shimohozumi, Ibaraki, Osaka 567-8680, Japan. [2]Laboratory for Molecular Design of Pharmaceutics, Faculty of Pharmaceutical Sciences, Hokkaido University, Kita-12, Nishi-6, Kita-Ku, Sapporo 060-0812, Japan. ✉e-mail: kazuki.hashiba@nitto.com; y_sato@pharm.hokudai.ac.jp

long-term stability of mRNA/LNPs at higher temperatures. To the best of our knowledge, this is the first study on the control of aldehyde production based on lipid structure.

In this study, we identified piperidine-based ionizable lipids that maintained mRNA/LNP activity even after refrigerated storage as liquid formulations. This study provides promising insights into enhancing the shelf-life of LNP-based mRNA from the viewpoint of lipid structure.

## Results

### Discovery of piperidine lipids
A library of 23 ionizable lipids with *N*-methyl piperidine head groups was designed for mRNA delivery (Fig. 1a). In earlier work on siRNA delivery, *N*-methyl piperidine was identified as one of the promising head groups[19]. In addition, heterocyclic amine-containing lipids were reported to elicit a strong immune response in mRNA/LNPs[20]. To enhance functional mRNA delivery, a variety of branched structures were introduced into our current lipid library. The intermediate containing the piperidine structure, synthesized from 6-bromo-1-hexanol, was esterified with branched tails to yield piperidine-ionizable lipids. These lipids were named CL15F m-n, where "m" and "n" are the main and side chain lengths, respectively (Fig. 1a). The final products were obtained by purification via both reverse- and normal-phase chromatography and identified using NMR and mass spectrometry.

To evaluate the potency of the CL15F lipid library, we compared it with the CL4F lipid library, which we developed previously[21] (Fig. 1b). LNPs were assembled by vigorously mixing ionizable lipid, cholesterol (Chol), 1,2-distearoyl-*sn*-glycero-3-phosphocholine (DSPC), and 1,2-dimyristoyl-*rac*-glycero-3-methoxypolyethylene glycol-2000 (DMG-PEG$_{2k}$) at a fixed molar ratio with mRNA using a microfluidic device (Fig. 1c). While LNPs containing CL15F 6-2, which had the lowest total carbon number, aggregated during the formation process owing to poor hydrophobic interactions, all other CL15F lipids were well formulated without any issues. All LNPs were positively charged in a pH-dependent manner. The apparent pKa values were between 6.24–7.15, which is ideal for mRNA delivery (Supplementary Fig. S1).

We first evaluated the in vitro functional delivery of firefly luciferase (FLuc) mRNA using CL15F LNPs. Most CL15F LNPs induced more intense bioluminescence than CL4F LNPs in HEK-293T cells (Fig. 1d, Supplementary Fig. S2a). In addition, some CL15F LNPs exhibited an efficacy comparable to that of Lipofectamine MessengerMAX, a reagent specifically optimized for mRNA transfection in vitro. To elucidate structure-activity relationships (SARs), each lipid structure was described by two parameters (total carbon number and symmetry of the tail). The impact of lipid structure on luciferase activity was visualized as a contour plot (Supplementary Fig. S2b), confirming that ionizable lipids with branching and longer tails can enhance the functional delivery of mRNA.

To examine the potential application of CL15F for vaccination, mice were immunized intramuscularly with prime and booster doses of ovalbumin (OVA) mRNA carrying LNPs (Supplementary Table S1). As positive control lipids, we used lipids that have been clinically successful, namely ALC-0315 and SM-102 (Fig. 1e). OVA antibody titers were comparable between CL15F, SM-102, and ALC-0315 LNPs (Fig. 1f, g). In addition, mice immunized with CL15F LNPs, particularly those derived from lipids with short tails, exhibited significantly stronger antigen-specific cellular responses than mice immunized with SM-102 and ALC-0315 LNPs (Fig. 1h, i). Compared with SM-102, CL15F 9-5 led to a 14-fold increase in IFN-γ spots.

### CL15F LNPs control the loss of functional delivery of mRNA at refrigeration temperature
The storage stability of mRNA/LNP formulations remains a significant challenge, particularly at refrigeration temperatures. We utilized the hEPO reporter system as a simple assay for evaluating the storage stability of mRNA/LNPs[22]. To compare the in vivo efficacy before and after mRNA/LNP storage, liver-targeted LNPs are desirable as they can induce hEPO levels sufficient for quantitative evaluation. We examined CL15F 12-10 and

CL15F 14-12 and compared them with CL4F 8-6, CL4F 10-4, SM-102, and ALC-0315.

To assess the in vivo efficacy of hEPO mRNA/LNPs, serum hEPO levels were quantified using ELISA following LNP administration at 0.25 mg/kg. hEPO secretion was confirmed in mice treated with fresh mRNA/LNPs, and this efficacy was comparable to that reported earlier[23,24] (Supplementary Fig. S3). We then investigated the in vivo efficacy of the LNPs stored at the given temperatures and time periods (Fig. 2a). The loss of hEPO expression in mRNA/LNP treatment was limited at -80 °C storage for up to 5 months (Fig. 2b); this is consistent with the knowledge that ultra-low temperature storage with cryoprotectant minimizes damage to LNPs and preserves mRNA potency[25]. CL15F LNPs maintained their in vivo activity after 5 months of storage even at 4 °C as a liquid formulation, whereas the activity of other LNPs decreased over time, with a half-life of approximately 2 months under consistent storage conditions (Fig. 2c). No in vivo toxicity was observed, including body weight change, and abnormal serum biochemistry parameters and histological scores at the consistent mRNA dose (Supplementary Fig. S4).

None of the tested LNPs showed obvious changes in their physicochemical properties or mRNA encapsulation (Supplementary Table S2). The integrity of each lipid component during storage was confirmed through high-performance liquid chromatography (HPLC) using a corona-charged aerosol detector (CAD)[26] (Fig. 2d, e, Supplementary Fig. S5).

### Aldehyde impurities are responsible for mRNA deactivation
To evaluate the differences in shelf-life between CL15F LNPs and the other LNPs, we focused on the interaction between mRNA and LNPs, as reported by Packer et al.[18]; oxidative impurities in ionizable lipids, *N*-oxides, and the resulting hydrophobic aldehyde impurities lead to deactivation of mRNA cargo owing to their potential reactivity with nucleosides.

To confirm this, we evaluated the relative amount of aldehyde impurities using a simple fluorescence-based microplate assay with 4-hydrazino-7-nitro-2,1,3-benzoxadiazole hydrazine (NBD-H), which reacts with carbonyl compounds and converts them into fluorescent hydrazones[27,28] (Fig. 3a). NBD-H emitted fluorescence only in the presence of aldehyde species under mild conditions (Supplementary Fig. S6). When ionizable lipids and helper lipids were incubated with NBD-H, the sample with CL15F and helper lipids emitted significantly lower fluorescence signals than the samples with CL4F and other ionizable lipids (Fig. 3b). Therefore, ionizable lipids are a major source of aldehyde impurities, whereas CL15F ionizable lipids yield minimal aldehyde species.

To identify the chemical structure of the aldehyde impurities originating from ionizable lipids, we performed LC-MS analysis on CL4F lipid samples labeled with 2,4-dinitrophenylhydrazine hydrochloride (DNPH). Carbonyl compounds reacted with DNPH to form stable hydrazones, which were detected using a UV detector; this enabled the detection of small amounts of aldehyde impurities in the samples. CL4F 10-8 and CL4F 11-5, which exhibited higher values in the NBD-H assay, were selected as samples. The absorbance at 360 nm in the UV chromatogram revealed the presence of DNPH-derivatized carbonyl compounds in each sample (Fig. 3c). In addition, mass spectrometry coupled with isotopic patterns allowed us to identify the peak of interest as a fatty aldehyde corresponding to each ionizable lipid with high reliability.

We assessed whether *N*-oxidized ionizable lipids are the main drivers of aldehyde production, as indicated by Packer et al.[18]. We synthesized *N*-oxidized CL4F lipids and incubated them in an acidic environment. *N*-oxidized CL4F lipids generated higher levels of aldehyde impurities than the non-oxidized parent lipids (Supplementary Fig. S7).

Therefore, aldehyde impurities were produced from CL4F lipids via *N*-oxidation (Fig. 3d). This finding is supported by the detection of dipropylamine and fatty secondary amines corresponding to *N*-oxidized CL4F lipids (Supplementary Fig. S8, S9). Among these degradation byproducts, fatty aldehydes possibly play a significant role in the loss of mRNA function, given their ability to localize within the LNP core and react with nucleobases (Supplementary Fig. S10). The logP values for propionaldehyde and

**Fig. 1 | Structure and efficacy of piperidine-based ionizable lipids. a** Synthetic route of branched ionizable lipids with piperidine, CL15F m-n. Chemical structure of the α-branched carboxylic acid used as a hydrophobic tail in this study. **b** Chemical structures of the previously developed ionizable lipids, CL4F m-n. **c** Simplistic illustration for LNP formation by mixing lipids and mRNA with a microfluidic device. The lipid molar ratio was fixed at ionizable lipid: cholesterol: DSPC: DMG-PEG$_{2k}$ = 50: 38.5: 10: 1.5. **d** Relative luciferase expression in HEK-293T cells 24 h after incubation with 100 ng/well FLuc mRNA carried in 10 types of CL4F LNPs, 21 types of CL15F LNPs, and Lipofectamine MessengerMAX. **e** Lipid structure of ALC-0315 (*left*) and SM-102 (*right*). **f** Timeline of the in vivo study to evaluate the potential application of CL15 LNPs for vaccine use (1 μg OVA mRNA per mouse, *n* = 4 biologically independent mice per group). **g** Anti-OVA IgG titers were determined by ELISA 3 weeks after a second intramuscular injection. Statistical significance was calculated by one-way ANOVA with Tukey's multiple comparisons test. **h**, **i** OVA-specific interferon-γ (IFN-γ) secreting cells were counted by enzyme-linked immunospot assay 1 week after a second intramuscular injection. Data were analyzed using one-way ANOVA with Dunnett's test for multiple comparison with ALC-0315. Mean ± SD.

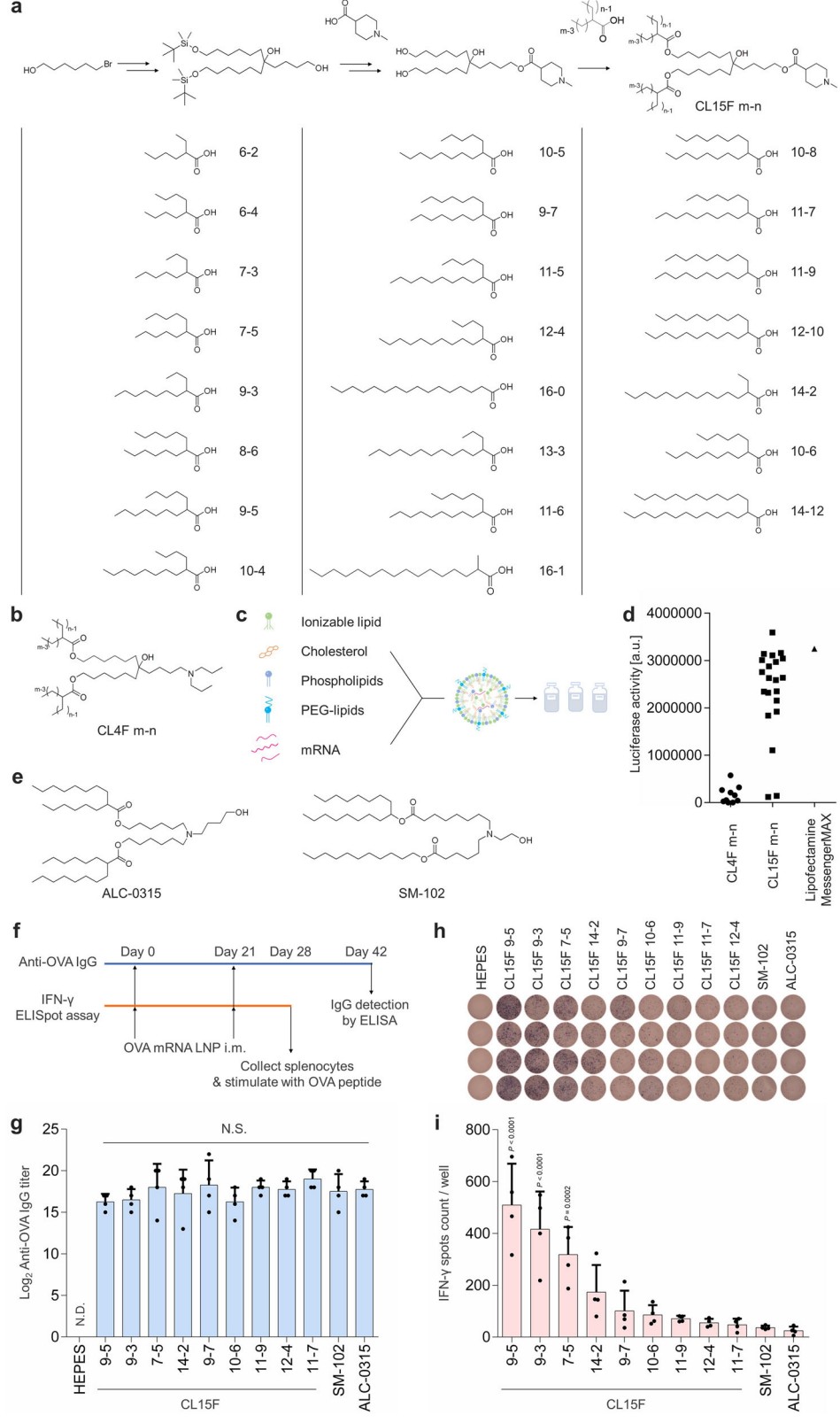

formaldehyde were 0.59 and 0.35, respectively. In the case of Spikevax, the volume ratio of lipids to water is approximately 0.4%. Therefore, most (98.5 ~ 99% or more) of the water-soluble aldehydes generated from *N*-oxides inside the LNP would possibly be distributed in the outer aqueous phase and be unlikely to react with mRNA.

## Adduct formation between aldehyde impurities and mRNA

To assess the potential impact of aldehyde impurities on mRNA function, we first investigated their reactivity with nucleosides. CL4F 11-5 ionizable lipids were incubated with each nucleoside, and the samples were analyzed using hydrophilic interaction chromatography (HILIC). The HILIC mode,

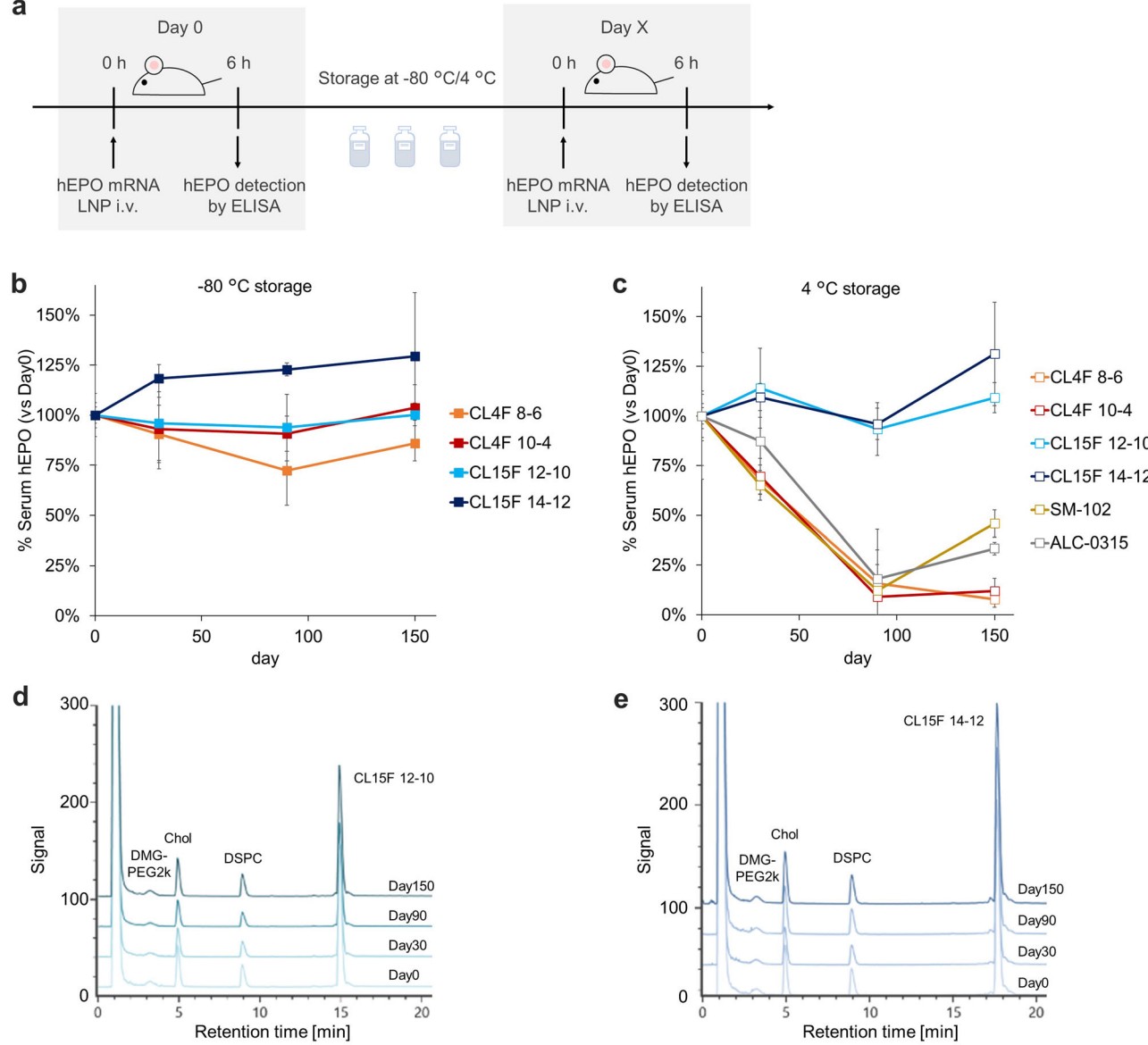

**Fig. 2 | Thermostability of mRNA/LNPs influenced by ionizable lipid structure.**
**a** Timeline of the in vivo study to evaluate storage stability of each LNP. In vivo assay was performed after 1-, 3- and 5-month storage of LNPs at each temperature ($n = 3$ biologically independent mice per group). **b** Relative hEPO expression (vs. day 0) 6 h after i.v. administration with 0.25 mg/kg hEPO mRNA encapsulated in LNPs. The in vivo efficacy of each mRNA/LNP was maintained for 5 months at −80 °C storage. **c** CL15F 12-10 and CL15F 14-12 enabled long-term storage of mRNA/LNP at 4 °C. **d** CAD chromatograms after 1-, 3-, and 5-month storage of CL15F 12-10 LNPs at 4 °C. **e** CAD chromatograms after 1-, 3-, and 5-month storage of CL15F 14-12 LNPs at 4 °C. Mean ± SD.

which is designed to retain small polar compounds using MS-friendly mobile phases, allows easier separation and identification of aldehyde adduct nucleosides[29]. A peak prior to each nucleoside peak was detected in the samples incubated with CL4F lipids, except for the guanosine-lipid conjugate (Fig. 4a–d). The mass spectra indicate that each earlier peak corresponds to nucleosides modified by fatty aldehydes derived from CL4F 11-5 (Supplementary Fig. S11). Under these conditions, the relative peak area derived from the adducts varied; this applied to the absence of lipid-guanosine adducts, which could be attributed to the difference in the reactivity and stability depending on the nucleobase type[30].

Next, we investigated the adduct formation between mRNA and fatty aldehyde impurities derived from ionizable lipids. After incubating FLuc mRNA with lipids as a simplified model system, the extracted mRNA was analyzed using reversed-phase ion pair chromatography. The mRNA complexed with ion-pair reagents showed increased hydrophobicity and enhanced retention in reversed-phase chromatography. mRNAs modified by

fatty compounds had increased hydrophobicity and were observed as late-eluting peaks. mRNA–lipid adducts were detected in preparations with CL4F lipids; however, the detection was minimal in those with CL15F lipids (Fig. 4e). To quantify adduct formation, the late-eluting peak areas were expressed as a relative percentage of the total peak area. Figure 4f shows how adduct formation proceeds over time. Adduct formation was detected at 1 h and further proceeded rapidly over time in the linear amine lipids including the CL4F group and ALC-0315 containing a high amount of fatty aldehyde. In contrast, limited amounts of adduct were observed in the CL15F group. To further demonstrate the role of the aldehyde impurities in adduct formation, a chemical scavenger was used. After the removal of aldehyde impurities from CL4F 11-9 lipids using polystyrene sulfonyl hydrazide with a catalytic amount of acetic acid, adduct formation with mRNA was analyzed. The adduct peak disappeared almost completely on aldehyde removal (Fig. 4g). In contrast, the removal of secondary amine impurities from CL4F 11-9 lipids using polystyrene methylisocyanate did not limit adduct formation.

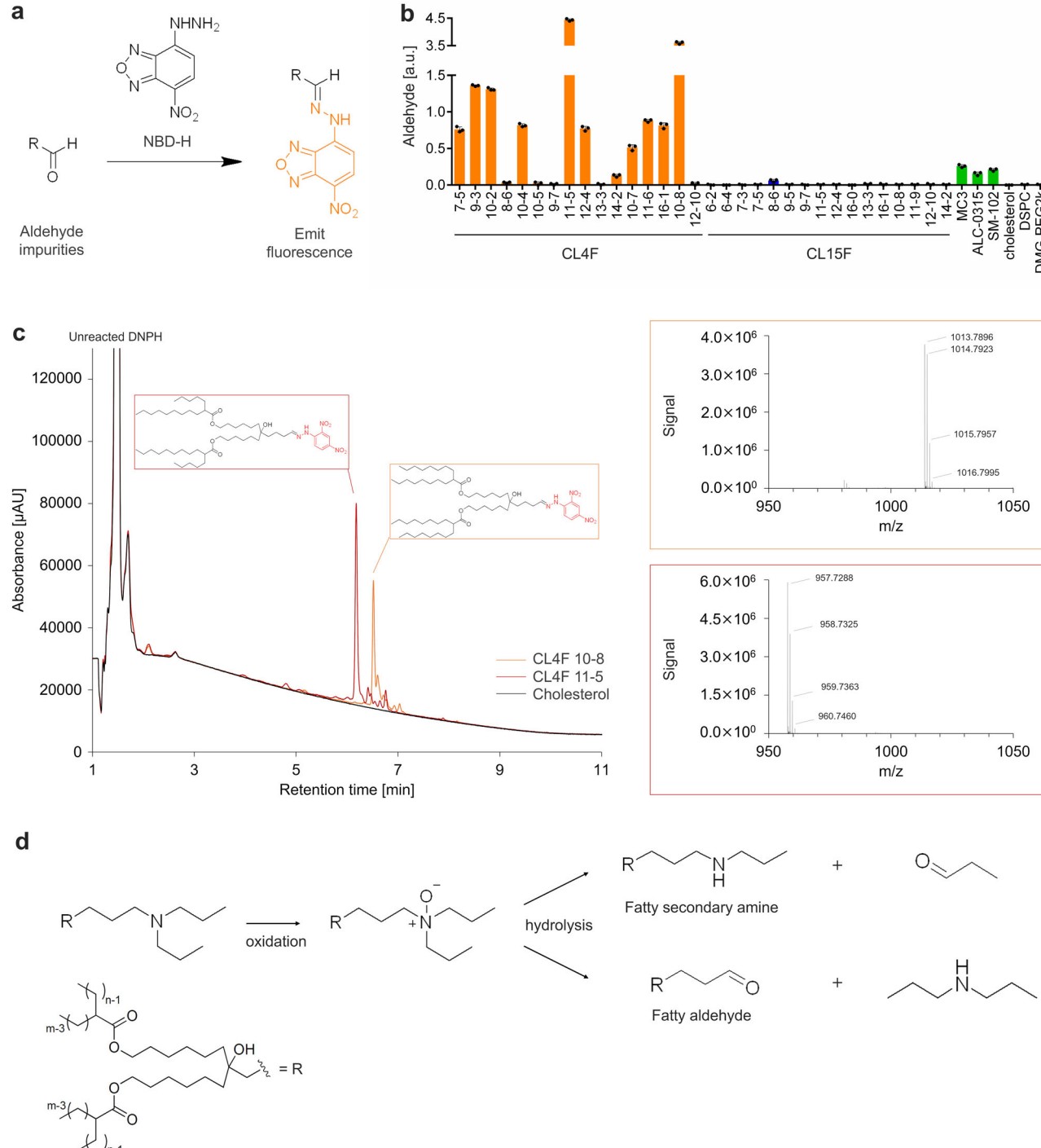

**Fig. 3 | Quantitative and qualitative analysis of aldehyde impurities. a** NBD-H reacts with aldehyde impurities in ionizable lipids to emit fluorescence. **b** The 38 types of ionizable lipids and other lipids were incubated with NBD-H for 60 min at 25 °C. After incubation, fluorescence intensity was compared; almost all CL15F lipids showed a small amount of aldehyde impurities compared to CL4F lipids and other ionizable lipids. Mean ± SD. **c** (*left*) DNPH-labeled aldehyde impurities were detected using a UV detector with reversed-phase ultra-high-performance liquid chromatography with QTOF mass spectrometry (UPLC-QTOF-MS); CL4F 10-8 (orange), CL4F 11-5 (red), and cholesterol (black) as the negative control. DNPH-

derivatized fatty aldehyde impurities eluted at 6.5 min with CL4F 10-8 and at 6.2 min with CL4F 11-5. (*right*) MS spectra of each peak in the range m/z 950 − 1050. The MS spectrum from the peak in DNPH-derivatized CL4F 10-8 corresponds to the exact mass (1013.7887) of fatty aldehyde generated from CL4F 10-8 (Upper right). The MS spectrum from the peak in DNPH-derivatized CL4F 11-5 corresponds to the exact mass (957.7261) of fatty aldehyde generated from CL4F 11-5 (Lower right). **d** The main source of aldehyde byproducts is CL4F lipids via *N*-oxidation. In particular, fatty aldehydes localize internally in LNPs and react with the mRNA encapsulated in LNPs to inactivate mRNA.

We hypothesized that the aldehyde impurities produced from typical lipids during the synthesis and/or LNP storage react with mRNA at 4 °C, resulting in the loss of hEPO expression, whereas CL15F produces minimal aldehyde byproducts, limiting adduct formation and maintaining its activity

at 4 °C. We performed an additional experiment to understand the relationship between aldehyde impurities and storage stability at 4 °C. The CL4F and CL15F LNPs carrying FLuc mRNA were formulated and stored at 4 °C in liquid form. On days 0 and 60, the bioluminescent signals in the liver

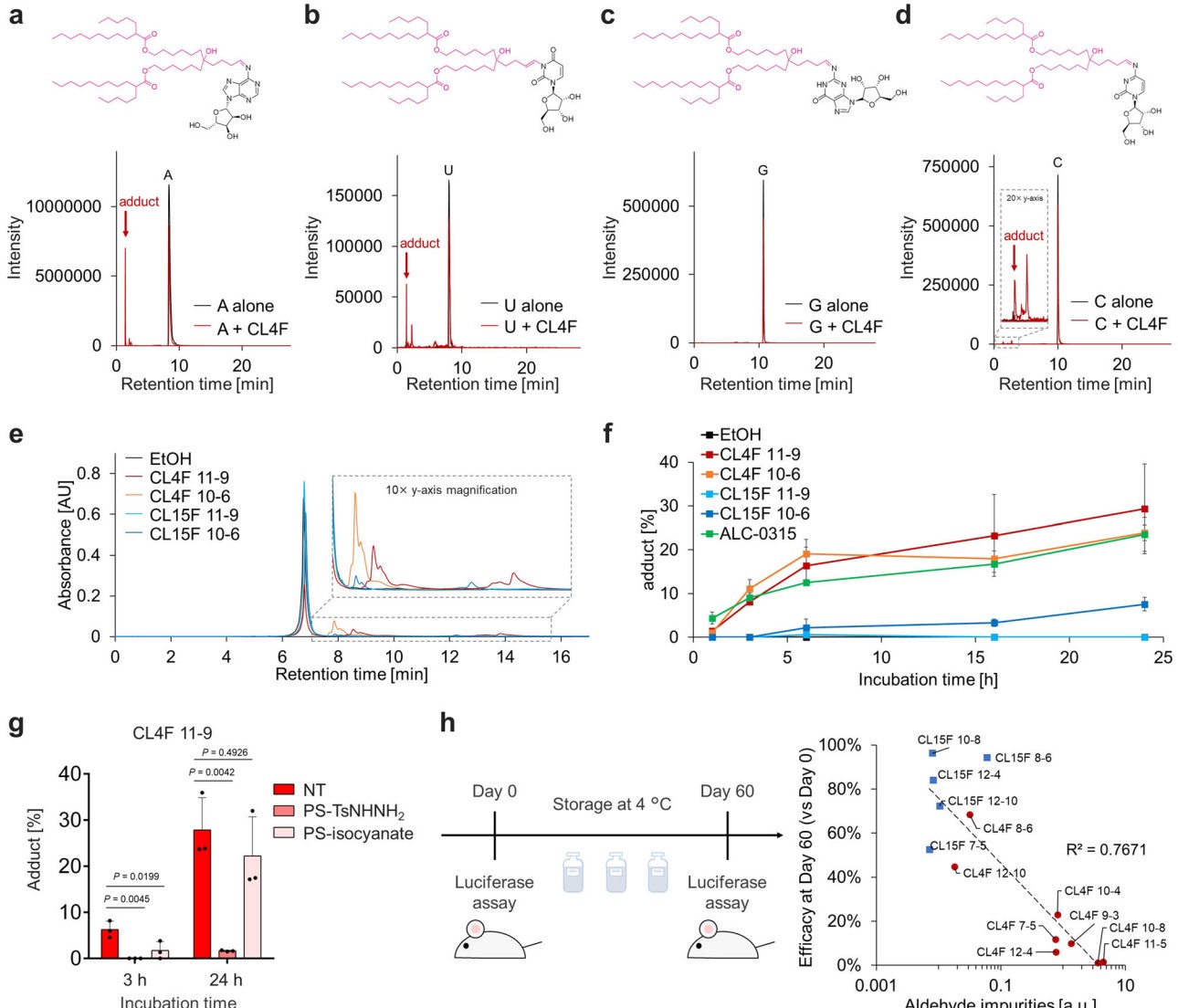

**Fig. 4 | Aldehyde impurities causing adduct formation with mRNA. a–d** CL4F 11-5 was incubated with each nucleoside (adenosine, uridine, guanosine, and cytidine) at 60 °C for 72 h. After incubation, each sample was analyzed using hydrophilic interaction chromatography with mass spectrometry. Overlay spectra of extracted ion chromatograms (EICs) of selected mass-to-charge ratios (m/z) corresponding to unmodified nucleosides and lipid-modified nucleosides in the sample incubated with (red) or without (black) CL4F lipids. The selected m/z for EICs include unmodified nucleosides (uridine; 8.0 min, adenosine; 8.4 min, cytidine; 10.0 min, guanosine; 10.7 min) and several lipid-modified nucleosides (1.4–2.8 min) except guanosine. **e** Twenty-four hours after incubation of FLuc mRNA with CL4F 10-6, CL4F 11-9, CL15F 10-6, and CL15F 11-9, mRNA was extracted using isopropanol precipitation and analyzed using reversed-phase ion pair chromatography. Unmodified mRNA eluted at 6.8 min; in contrast, lipid-modified mRNA eluted later at 7.5−15.5 min. 10× Y-axis zoom of late-eluting peaks is shown in the inset. **f** Percentage of adduct formation was calculated from the late-eluting peak areas relative to the total peak area. **g** Percentage of adduct formation between mRNA and CL4F 11-9 after removal of aldehyde or secondary amine compounds from CL4F 11-9 using PS-TsNHNH₂ or PS-isocyanate, respectively. Most of the adduct peaks diminished when the aldehydes were removed but not when amines were removed, indicating that the aldehydes corresponding to the ionizable lipids are responsible for adduct formation. Data were analyzed using one-way ANOVA with Dunnett's test for multiple comparison with NT. **h** Scatter plot showing the relationship between the reduction in bioluminescence after storage for 60 days at 4 °C and the relative amount of aldehyde impurities at day 0 in the ionizable lipids. The bioluminescence from the liver was captured using an in vivo imaging system (IVIS) 6 h after i.v. administration of FLuc mRNA/LNPs. Relative aldehyde impurities were estimated using NBD-H assay. The eight CL4F LNPs are indicated using circles (red) and the five CL15F LNPs are indicated using squares (blue). Their coefficient of determination was 0.767 (0.1 mg/kg FLuc mRNA per mouse, $n$ = 3 biologically independent mice per group). Mean ± SD.

tissue were measured 6 h after i.v. administration, and the loss of mRNA activity 2 months after storage was calculated. All LNPs maintained particle size, polydispersity index (PDI), and encapsulation efficiency during the 3-month storage at 4 °C (Supplementary Fig. S12). However, the reduction in mRNA delivery efficiency after 2 months of storage at 4 °C varied markedly depending on the ionizable lipid; this ranged between 96.3% and 1.0% (Fig. 4h). Consistent with previous results, CL15F LNPs showed enhanced storage stability compared with CL4F LNP. Figure 4h illustrates the relationship between the reduction in mRNA activity and the relative

amount of aldehyde impurities indicated by NBD-H. Therefore, the aldehyde impurities derived from ionizable lipids were an important factor for the long-term storage of mRNA/LNPs at 4 °C and could be controlled by the chemical structure of the lipid.

## Oxidative impurities in CL15F lipids
We hypothesized that the unique molecular structure of the current ionizable lipid might be a key factor contributing to the reduced aldehyde impurities and enhanced stability at 4 °C. Specifically, the heterocyclic

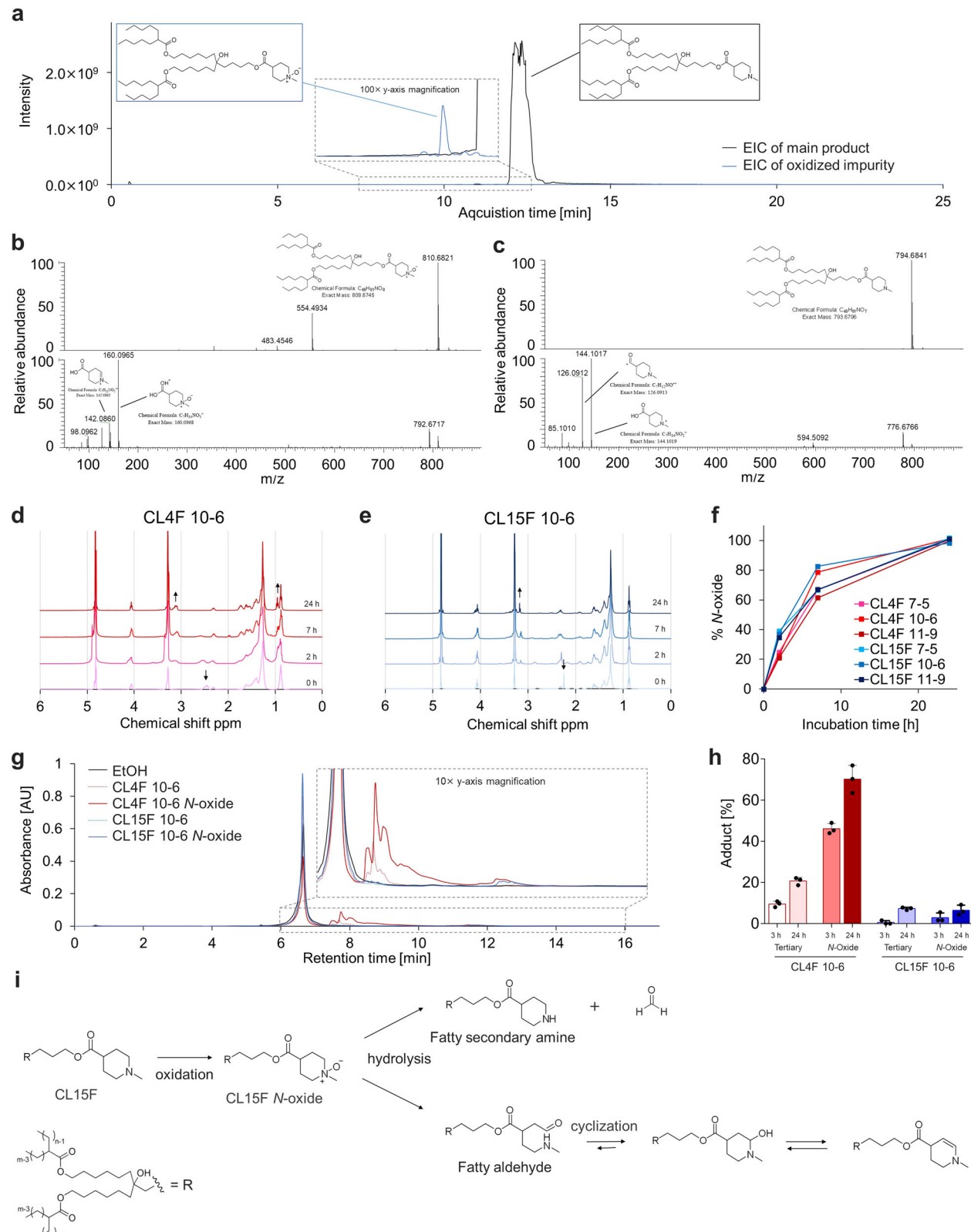

amine structure of CL15F could play a vital role in preventing the production of aldehydes from *N*-oxide moieties, which are known precursors of aldehyde impurities.

We investigated the oxidative impurities in the CL15F lipids using high-resolution MS/MS. The EIC of the selected m/z indicated that the peak eluted at 11.0 min, corresponding to the oxidized form of the lipid (Fig. 5a). To further elucidate the oxidized structure, the fragmentation spectra were reviewed to determine the position of oxygen incorporation. The CID MS$^n$ spectra were insufficient to localize modifications on the molecule (Supplementary Fig. S13); therefore, we used higher-energy collisional dissociation (HCD), which tends to produce fragment ions of low m/z compared to CID MS$^n$ [31]. There were unique fragment ions, such as oxidized cyclic amine and cyclic imine, present only in the MS/MS spectra from the oxidized form, confirming that oxidation was localized to the tertiary amine (Fig. 5b, c).

**Fig. 5 | Limited adduct formation by piperidine-based lipids and their oxidized lipids. a** CL15F 7-5 lipid analyzed using UPLC with tandem MS spectrometry. EICs of selected m/z corresponding to CL15F 7-5 (black) and oxidized species (blue). **b** MS (upper panel) and high-energy MS/MS (lower panel) spectrum of the oxidized species in CL15F 7-5 were acquired using the Orbitrap mass analyzer. The fragment ion of m/z 160.0965 corresponds to the exact mass of protonated 1-methylpiperidine-4-carboxylic acid N-oxide. The fragment ion of m/z 142.0860 corresponds to the loss of water, confirming the localization of oxidation in tertiary amine[43]. **c** MS (upper panel) and high-energy MS/MS (lower panel) spectrum of the main species in CL15F 7-5. Fragment ions did not contain the above-mentioned N-oxide-derived peaks, but only pure tertiary amine-derived peaks such as 144.1017 and 126.1912. **d**, **e** Oxidation of CL4F 10-6 and CL15F 10-6 by hydrogen peroxide and monitoring oxidation kinetics using ¹H-NMR. As the oxidation progressed, the peak at 2.40 − 2.60 ppm corresponded to the methylene group of CL4F 10-6, and the peak at 2.25−2.30 ppm corresponding to the methyl group attached to the nitrogen

atom of CL15F 10-6 disappeared. The peak at 3.05−3.18 ppm corresponding to methyl/methylene group attached to the oxidized nitrogen atom appeared, as indicated by the arrows. **f** The oxidation kinetics of three types of CL4F and CL15F lipids were estimated from the relative peak area specific to N-oxide. **g** Three hours after incubation of FLuc mRNA with CL4F 10-6, CL15F 10-6, and corresponding N-oxides, the extracted mRNA was analyzed using reversed-phase ion pair chromatography. Mean ± SD. **h** Three or Twenty-four hours after incubation of FLuc mRNA with CL4F 10-6, N-oxidized CL4F 10-6, CL15F 10-6, and N-oxidized CL15F 10-6, extracted mRNA was analyzed using reversed-phase ion pair chromatography and then the percentage of adduct formation was calculated. Oxidation of CL4F lipids accelerated adduct formation but the oxidation of CL15F lipids did not influence adduct formation. **i** N-oxidation proceeded even with CL15F lipids, but adduct formation was limited compared to that with CL4F lipids. This could be because hydrolysis was unlikely to proceed or because a cyclic imine can be formed if an amine and an aldehyde are present on the same molecule.

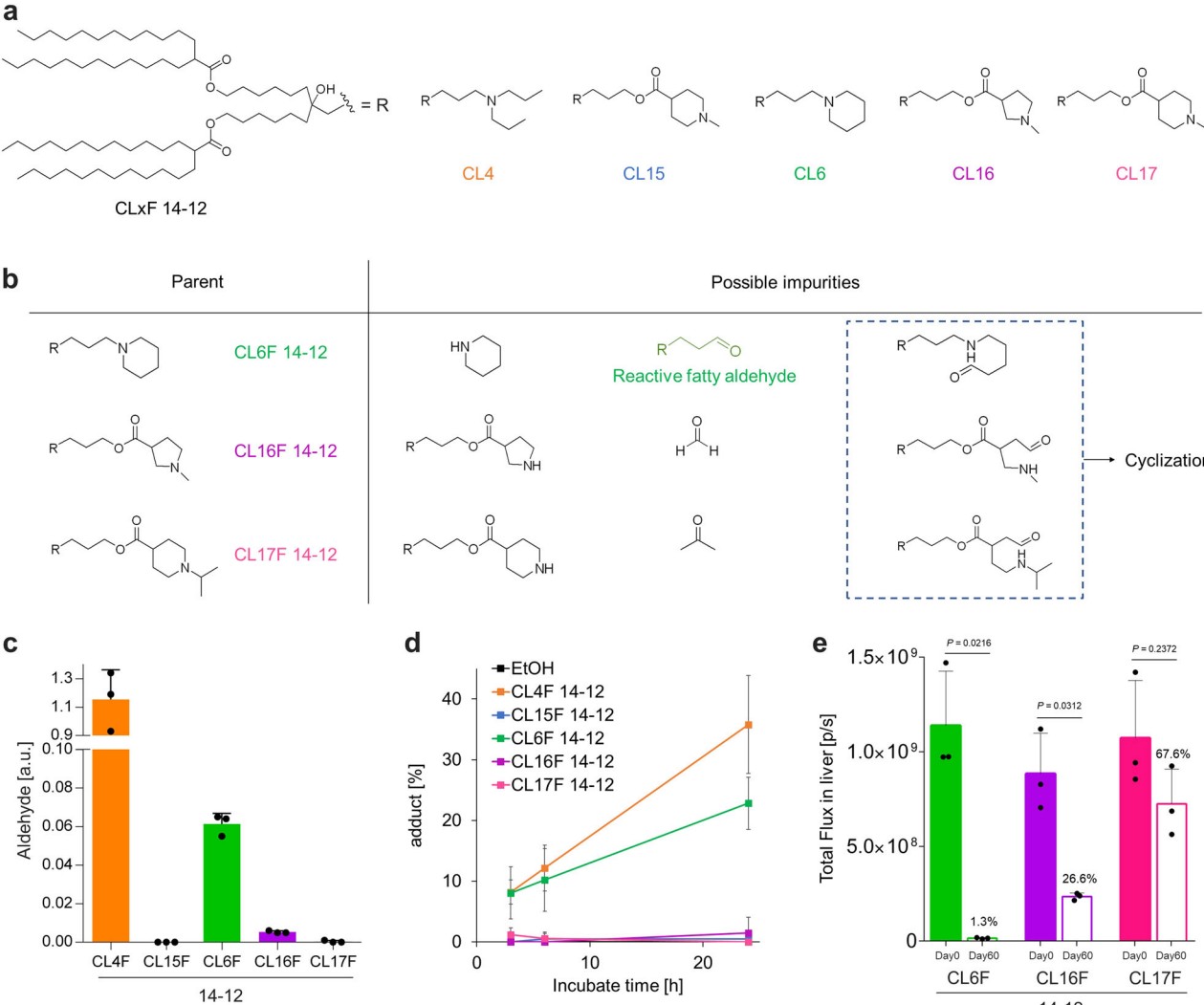

**Fig. 6 | Limited mRNA–lipid adduct formation by cyclic amine lipids. a** Lipid structure was expanded with three different cyclic amine moieties, CL6F 14-12, CL16F 14-12, and CL17F 14-12. **b** Predicted impurity structures when each ionizable lipid was oxidized and subsequently hydrolyzed. **c**) Relative amount of aldehyde impurities in additional lipids was evaluated using NBD-H. **d** After incubation of FLuc mRNA and lipids in the simplified model, adduct formation was estimated using reversed-phase

ion pair chromatography. **e** Fluc mRNA/LNPs were formulated using CL6F14-12, CL16F 14-12, and CL17F 14-12. For fresh LNPs and LNPs stored for 60 days at 4 °C, the bioluminescence was captured using IVIS 6 h after i.v. administration of LNPs. The relative change in total luminescence from the liver is shown in the graph (0.1 mg/kg FLuc mRNA per mouse, n = 3 biologically independent mice per group). Data were analyzed using the paired Student's t-test. Mean ± SD.

Next, we compared the oxidation kinetics of the CL15F and CL4F lipids. Four equivalents of hydrogen peroxide were added to the lipids and stirred at 50 °C. The oxidation kinetics were monitored using ¹H-NMR (Fig. 5d, e, Supplementary Fig. S14, S15).

The relative peak area unique to N-oxide was used to determine the progression of oxidation. No significant differences were observed in the oxidation kinetics of piperidine and linear amine lipids (Fig. 5f).

Next, we investigated whether the piperidine structure of the CL15F lipids prevented the formation of aldehydes from the *N*-oxide moiety. After incubating tertiary amine lipids or *N*-oxidized lipids with mRNA, lipid–mRNA adduct formation was quantified using reversed-phase ion pair chromatography. The oxidation of CL4F lipids accelerated adduct formation, whereas that of CL15F lipids did not (Fig. 5g, h). To further support these data, we investigated the relationship between *N*-oxide species and aldehyde impurities in ionizable lipids. CL4F *N*-oxide contributed to an increase in the aldehyde byproducts, whereas CL15F *N*-oxide did not (Supplementary Fig. S16).

Therefore, we propose a possible mechanism underlying the minimal aldehyde impurities detected with the CL15F lipids (Fig. 5i). *N*-oxidation proceeded evenly in CL15F and CL4F lipids. However, hydrolysis was hindered, or cyclic imines were immediately formed via a hemiaminal, that was produced by the intermolecular reaction between amines and aldehydes. Consequently, CL15F lipids generated minimal fatty reactive aldehydes, which were likely to be located in the LNP core, resulting in the enhanced storage stability of mRNA/LNPs.

### Impact of the heterocyclic structure on mRNA–lipid adduct formation

To further investigate the role of lipid structure in aldehyde generation, adduct formation, and storage stability, we synthesized additional lipids with three different cyclic amine moieties as the head structure (Fig. 6a). We selected a long and symmetrical tail, 14-12, with the most potential to physically stabilize LNPs[21] to rule out the possibility that LNPs may physicochemically deteriorate. From the above possible pathways (Fig. 5i), CL16F 14-12 and CL17F 14-12 do not generate reactive fatty aldehydes, even if oxidation and hydrolysis occur, because they can form 5- or 6-membered cyclic imines via intramolecular aldehyde–amine reactions (Fig. 6b). In contrast, CL6F 14-12, although heterocyclic, has the potential to deactivate mRNA because it has a nitrogen atom directly bonded to the scaffold, which can generate fatty byproducts containing only aldehyde groups as well as linear amine lipids.

CL16F 14-12 and CL17F 14-12 showed remarkable reductions in aldehyde impurities (Fig. 6c). In addition, they limited mRNA–lipid adduct formation in a simplified model system (Fig. 6d). In contrast, CL6F 14-12 included aldehyde impurities and promoted adduct formation with mRNA. These findings support the previously described mechanism (Fig. 5i).

FLuc mRNA/LNPs were prepared from three lipids, using the same composition and process as described previously (Fig. 4h). We compared the physical properties and in vivo mRNA expression of fresh LNPs and those stored at 4 °C for 2 months. While the particle size, potential, PDI, and encapsulation efficiency remained similar during storage (Supplementary Table S3), the bioluminescence was reduced following the intravenous injection of mRNA/LNPs; this was dependent on the ionizable lipid used (Fig. 6e). CL17F 14-12 LNPs exhibited a storage stability similar to that of CL15F LNPs, at 4 °C (Fig. 4h). In contrast, CL6F 14-12 and CL16F 14-12 showed significant mRNA deactivation during storage at 4 °C. This in vivo result for CL6F and CL17F was consistent with the relative aldehyde levels and adduct formation described earlier; this was not the case with CL16F. Therefore, minimizing aldehydes is necessary, but it is not sufficient to enhance the thermostability of mRNA/LNPs (see "Discussion"). Further studies are underway to evaluate mRNA integrity.

These findings highlight the importance of lipid structure in minimizing aldehyde impurities, preventing adduct formation, and enhancing the storage stability of mRNA/LNPs at 4 °C. The CL15F and CL17F lipids improved storage stability.

### Discussion

mRNA is thermally labile. To preserve its shelf-life, mRNA–LNP vaccines for the COVID-19 pandemic were distributed under ultra-cold storage conditions. Lyophilization can be employed to improve thermostability. Reducing the water from the formulation can suppress mRNA hydrolysis; however, scientists have opined that LNP size is likely to increase following

reconstitution. Some reports indicate that mRNA/LNPs can be stored at refrigerated temperatures as liquid formulations[22,32]. Unfortunately, these studies do not provide detailed mechanisms. A mechanism to govern the thermostability of mRNA-LNPs has recently been proposed. Fatty aldehyde impurities produced via the oxidation and hydrolysis of ionizable lipids form mRNA–lipid adducts and inhibit mRNA translation[18]. Owing to this, we hypothesized that aldehyde impurity generation and adduct formation could be suppressed through chemical engineering of lipid structures. In this study, we synthesized a lipid library with *N*-methyl piperidine head groups. From studies comparing in vivo mRNA delivery before and after storage, we demonstrated that piperidine-based lipids enabled long-term storage of mRNA/LNPs at 4 °C. To provide a better understanding of the underlying mechanism of this discovery, we performed HPLC analysis and additional lipid synthesis. Since C-N bonds in ionizable lipids are cleaved by oxidation and hydrolysis, the generated aldehyde structure depends on the position of the nitrogen atom in the hetero ring of the ionizable lipid. Both *N*-methylpiperidine CL15F lipid and the additionally synthesized CL6F lipid have heterocyclic amines; however, CL6F lipid, which has a scaffold directly bound to the nitrogen atom, yielded fatty aldehydes, formed undesired adducts with mRNA, and significantly decreased mRNA efficacy during storage. Based on our observation, it is plausible that impurified fatty aldehyde generation occurs from lipid structures such as the recently discovered piperazine derivatized lipids[33–35], but this needs further experimental verification in the future. Nevertheless, this study revealed that amine moieties of ionizable lipids play a vital role in controlling reactive aldehyde generation, mRNA–lipid adduct formation, and loss of mRNA function during the storage of mRNA/LNPs. Additionally, compared with ALC-0315 and SM-102, these lipids can significantly increase IFN-γ secretion, suggesting their potential utility in vaccine production for cancer prevention and treatment, particularly where cellular immunity is important. This result is consistent with the report that lipids containing heterocyclic groups induce a robust immune response compared with those containing linear tertiary amines[20].

Generally, the lipid and LNP manufacturing processes should be tightly controlled to minimize impurities that deactivate mRNAs, including *N*-oxide and its degraded form, reactive fatty aldehyde. However, the complete elimination of *N*-oxides is challenging. Considering the COVID-19 mRNA vaccine (~4000 bases, N/P = ~ 6), ionizable lipids exist in a molar ratio 24,000 times higher than that of mRNA. Even if *N*-oxides were reduced to 0.1% through strict quality control, there would still exist 24 equivalents of *N*-oxides compared to the amount of mRNA. There is a need to fulfill the increasing interest in functional and long mRNA molecules, such as self-replicating RNA, to improve therapeutic outcomes. However, enhancing the thermostability of mRNA–LNP systems is difficult with long mRNA because it has a high tendency to lose its functional properties. Therefore, the present approach, which suppresses reactive aldehyde generation and adduct formation even in the presence of *N*-oxides holds significant importance. This approach also allows for combinations with other methods, such as the incorporation of a tris buffer to capture reactive aldehydes[36]. The resulting thermally stable LNPs would facilitate worldwide distribution of mRNA-related therapeutic agents.

Our study has some limitations. First, accurately estimating the impact of generated small aldehyde impurities on mRNA was difficult. The small aldehyde itself can be detected by HPLC analysis after DNPH labeling; however, it was problematic to detect small aldehyde–mRNA adducts, as changes in hydrophobicity are minimal when small aldehydes react with mRNA. Second, three linear and four cyclic lipids were investigated in this study. Nonetheless, further studies are required to assess the generality of this proposal. Third, the sequence (e.g. GC-content) and modification patterns that would contribute to double-strand formation can affect adduct formation efficiency since there is a small reaction between dsRNA and aldehydes[37,38]. It is known that mRNAs inside LNPs have more restricted higher-order structures than free mRNAs[39,40]. Therefore, ionizable lipid structure, lipid composition, and the LNP preparation process are likely to change the higher-order structure of mRNA and affect adduct formation.

Currently, it is difficult to accurately estimate the impact of these factors and simply discuss the reactivity of each nucleoside in LNPs, but is important to design formulations with these factors in mind in the future. Fourth, the loss of mRNA function can occur not only due to the formation of mRNA–lipid adducts but also because of in-line hydrolysis[41]. Base catalysis by the amine headgroup of lipids may facilitate nucleophilic attack of the ribose 2′-hydroxyl group on the phosphorus atom and accelerate backbone cleavage[42]. mRNA conformations, including the secondary structure, which may be altered by lipid conformation and pH, can potentially affect the hydrolytic kinetics of mRNA[11,22,39]. We observed a decrease in efficacy that cannot be solely attributed to aldehyde impurities. CL16F, which limited adduct formation with mRNA, had low efficacy after storage; CL15F LNPs exhibited different storage stabilities depending on the lipid tail structure. We are currently performing analysis to evaluate and ensure the integrity of the mRNA.

Despite these limitations, the findings presented in this study hold several important implications. This study underscores the crucial role that the amine structure of ionizable lipids plays in aldehyde impurity generation, enabling the storage of mRNA/LNP formulations at temperatures higher than cryogenic conditions. Storing mRNA/LNP formulations at higher temperatures can potentially simplify distribution and storage logistics. This has significant implications for cost-effectiveness and expanding the accessibility of mRNA therapeutics to regions with limited cryogenic capabilities. Successful identification of piperidine-based ionizable lipids with improved stability characteristics paves the way for further research directions. Furthermore, deeper analysis is needed for a comprehensive understanding of how lipid structures influence aldehyde impurity generation and mRNA stability. The strategic designing of ionizable lipids, lipid composition, and manufacturing processes that comprehensively consider the above perspectives could resolve the storage stability issues of mRNA/LNPs.

## Methods
### Ionizable lipid synthesis
In general, branched fatty acid (2.4 mmol) was added to 5,11-duhydroxy-5-(6-hydroxyhexyl)undecyl 1-methylpiperidine-4-carboxylate (430 mg, 1.0 mmol) in anhydrous DCM (5 mL). DMAP (12.2 mg, 0.1 mmol) and EDCI-HCl (576 mg, 3.0 mmol) were added to the mixture and the reaction mixture was stirred at 25 °C overnight. After evaporation of the solvent, the residue was suspended in ethyl acetate, washed with a 0.5 N NaOH solution, and then brine. The organic phase was then dried over $Na_2SO_4$. Evaporation of the solvent yielded a crude yellow oily residue. The residue was purified using flash chromatography [ODS, $H_2O$ (0.1% trifluoroacetic acid)/acetonitrile:isopropanol = 50:50 (0.1% TFA)] and ($SiO_2$, DCM/MeOH). The full synthesis protocol and representative NMR spectra are included in the supporting information.

### Formulation of mRNA-encapsulated LNPs
mRNA was loaded into the LNPs using microfluidic mixing methods. mRNA used in this study was modified with 5-methoxyuridine and was capped using CleanCap; it was purchased from TriLink BioTechnologies (San Diego, CA, USA). Briefly, an 8 mM lipid ethanol solution and citrate buffer (50 mM, pH 3.5) containing mRNA were rapidly mixed using a NanoAssemblr Benchtop (Precision Nanosystems, Vancouver, BC, Canada) at an N/P ratio of 10. The resulting LNP solutions were neutralized through buffer exchange with 20 mM HEPES buffer (9% sucrose, pH 7.45) using an ultrafiltration device (Amicon Ultra-15, MWCO 100 kDa; Millipore, Billerica, MA, USA). Sucrose was used as a cryoprotectant. The final LNPs were stored in liquid form at 4 °C or frozen at -80 °C under a nitrogen atmosphere.

### mRNA/LNP characterization
mRNA/LNPs were diluted in D-PBS (-) to measure the particle size, PDI, and zeta potential using a Zetasizer Nano ZSP instrument (Malvern Instruments, Worcestershire, UK).

To quantify the total mRNA concentration and encapsulation efficiency of mRNA/LNPs, the Ribogreen assay was performed. Briefly, LNPs were diluted in 10 mM HEPES buffer at pH 7.4 containing RiboGreen in the presence or absence of 0.1 w/v% Triton X-100. Fluorescence was measured by Varioskan LUX Multimode Microplate Reader (Thermo Fisher Scientific, MA, USA) at an excitation wavelength of 500 nm and an emission wavelength of 525 nm in black 96-well plates at a total volume of 200 μL. The mRNA encapsulation efficiency was calculated by comparing mRNA concentration in the presence and absence of Triton X-100.

To measure the apparent pKa of mRNA/LNPs, the TNS assay was performed. Briefly, 6-p-toluidino-2-naphthalenesulfonic acid (TNS) was diluted with each buffer and the pH ranged from 3.5–9.5 in 0.5 increments to a final concentration of 0.8 μM. The LNP was diluted with saline to 1 mM as the total lipid concentration. In black 96-well plates, 12 μL of LNP and 188 μL of TNS solutions for each pH were mixed. The fluorescence intensity was measured using a Varioskan LUX Multimode Microplate Reader at an excitation wavelength of 321 nm and an emission wavelength of 447 nm in black 96-well plates at a total volume of 200 μL. The apparent pKa was calculated as the pH giving rise to the half-maximal fluorescent intensity.

### HPLC/CAD analysis for lipid integrity
hEPO mRNA/LNPs were diluted 100-fold with 70% isopropanol, and the lipid integrity was analyzed using a Waters BioAccord System with a charged aerosol detector (CAD). Separation was carried out using an ACQUITY UPLC BEH C18 Column (130 Å, 1.7 μm, 2.1 mm, 100 mm) and a gradient of 80–100% isopropanol/acetonitrile (62:33) in water with 5 mM ammonium acetate over 15 min and held at 100% isopropanol/acetonitrile (62:33) with 5 mM ammonium acetate for 5 min at 0.3 mL/min. The injection volume was 2.0 or 10.0 μL and the column temperature was 60 °C.

### Plate-reader-based fluorescence assay for lipid impurity measurement
NBD-H (4-hydrazino-7-nitro-2,1,3-benzoxadiazole hydrazine) was used for detecting aldehyde impurities in the lipids. Under mild conditions, NBD-H reacts with the aldehyde species to emit fluorescence; 250 μM NBD-H acetonitrile solution containing 0.025% trifluoroacetic acid (TFA) was prepared; 285 μL of NBD-H solution was added to 15 μL of 40 mM lipid in ethanol and incubated for 60 min at 25 °C. After incubation, fluorescence intensity was measured using a Varioskan LUX Multimode Microplate Reader at an excitation wavelength of 470 nm and an emission wavelength of 550 nm in black 96-well plates at a total volume of 200 μL.

### Determination of aldehyde impurities in lipids by DNPH derivatization and LC/MS detection
For aldehyde or ketone labeling, an acidified DNPH solution was prepared using 50 mg of 2,4-dinitrophenylhydrazine hydrochloride (Fujifilm Wako Pure Chemical, Japan), 90 mL of ethanol, 8 mL of ultrapure water, and 2 mL of hydrochloric acid. The lipid sample was solubilized in 1 mL ethanol, added to 1 mL DNPH solution, and incubated for 30 min at 45 °C for derivatization. The sample was cooled on ice, and it was analyzed using HPLC.

Chromatographic analysis was performed using the Acquity UPLC system (Waters, Milford, MA, USA) with an Acquity UPLC BEH C4 analytical column (100 mm × 2.1 mm, 1.7 μm, Waters) at a column temperature of 50 °C. Mobile phase A was 10 mM ammonium acetate, and mobile phase B was THF/acetonitrile (7:3). Separation was accomplished through a step-gradient with a 7.0-min gradient from 55 to 95% B and a hold at 95% B, delivered at a flow rate of 0.2 mL/min with an injection volume of 2 μL. The labeled aldehyde was detected using UV irradiation at 360 nm. Full MS scans (ESI-negative) were acquired using a Xevo G2-XS QTOF over an m/z range of 50–2000. High-resolution mass spectrometry, along with isotope patterns, allowed the determination of the precise chemical composition of the DNPH-labeled compounds with high reliability.

### Impurity structure analysis using high-resolution MS/MS
The structure of the impurity was analyzed using an UltiMate3000 instrument (Thermo Fisher Scientific) with an ACQUITY UPLC BEH C18

Column (130 Å, 1.7 μm, 2.1 mm, 100 mm) and a gradient of 50–95% iso-propanol/acetonitrile (62:33) in water with 5 mM ammonium acetate over 10 min and held at 95% isopropanol/acetonitrile (62:33) in water with 5 mM ammonium acetate for 10 min at 0.3 mL/min. The injection volume was 1.0 or 0.2 μL, and the column temperature was 60 °C. Mass spectral data were acquired using an LTQ Orbitrap XL in the positive electrospray ionization (ESI) mode. The electrospray voltage was 3.0 kV and the ion transfer capillary temperature was 350 °C. Full MS scans were acquired using an Orbitrap mass analyzer over the m/z range of 100–2000 with a resolution of 60,000. The selected precursor ion peaks were fragmented in the HCD collision cell with a normalized collision energy of 35%, and the tandem mass spectrum was acquired using an Orbitrap mass analyzer with a resolution of 30000.

### Lipid-modified nucleoside analysis using HILIC with mass detection

For monitoring the modification of nucleosides with lipid-derived impurities in a simplified system, incubated (60 °C, 72 h) mixtures containing 4 mM lipid ethanol solution, 80 μg/mL nucleosides in THF: ethanol (1:1) containing 1 mM sodium hydroxide, and acetic acid at a ratio of 50:50:4 were used as samples. Chromatographic analysis was performed using the Acquity UPLC system (Waters) with an Acquity UPLC BEH Amide analytical column (100 mm × 2.1 mm, 1.7 μm, Waters) at a column temperature of 40 °C. Mobile phase A was 10 mM ammonium formate, and mobile phase B was acetonitrile. Separation was accomplished using a step gradient with a hold at 98% B for 3 min and a 15.0-min gradient from 98 to 50% B and hold at 50% B, delivered at a flow rate of 0.2 mL/min with an injection volume of 5 μL. Each nucleoside was monitored under UV light at 260 nm. Full MS scans (ESI-positive) were acquired using Waters ZQ2000 over an m/z range of 50–2000 with a 30 V cone voltage.

### mRNA analysis using reversed-phase ion pair chromatography

Isopropanol precipitation was used to extract mRNA from the lipid mixture and LNPs. Specifically, 900 μL of isopropanol containing 60 mM ammonium acetate was added to 100 μL of the sample, vortexed, and centrifuged at 14000 g for 15 min at 4 °C. The supernatant was discarded; and the pellet was washed with 1000 μL of isopropanol, centrifuged at 4 °C, and vacuum dried. For monitoring the adduct formation of mRNA and lipid-derived impurities in a simplified system, incubated (25 °C, 700 rpm) mixtures containing 4 mM lipid ethanol solution and 0.135 mg/mL mRNA in 50 mM sodium acetate (pH 5.3) at a ratio of 1:3 were used as samples. The mRNA pellet was resuspended in ultrapure water for liquid chromatography-mass spectrometry. The mRNA solutions were diluted to 50 ng/μL and the concentration and purity were measured using NanoDrop One.

Chromatographic analysis was performed using an Arc HPLC system (Waters) with an DNAPac RP analytical column (50 mm × 2.1 mm, 4 μm, Thermo Fisher Scientific) at a column temperature of 65 °C. Mobile phase A comprised 50 mM dibutylammonium acetate (TCI, Japan) and 100 mM triethylammonium acetate (Fujifilm Wako Pure Chemical). Mobile phase B consisted of 50% acetonitrile, 50 mM dibutylammonium acetate, and 100 mM triethylammonium acetate. Separation was accomplished using a step gradient with an initial 0.75-min hold at 25% B, a 0.5-min gradient from 25 to 45% B, a 9.25-min gradient from 45 to 100% B, and a hold at 100% B, delivered at a flow rate of 0.35 mL/min with an injection volume of 10 μL (approximately 2 μg of mRNA). The mRNA was detected using UV light at 260 nm.

### Removal of aldehyde or amine impurities using scavenger resin

The lipids containing tertiary amines (0.01 mmol) were dissolved in dichloromethane (1.0 mL). To eliminate aldehyde impurities, PS-TsNHNH$_2$ (Biotage, Uppsala, Sweden) (0.03 mmol) and acetic acid (50 μL) were added, and the mixture was vigorously stirred at 40 °C for 3 h. To eliminate amine impurities, PS-isocyanate (Biotage) (0.06 mmol) was added, and the mixture was vigorously stirred at 40 °C for 24 h. The reaction mixture was filtered through Celite. The solvent and acetic acid were

removed *in vacuo* to obtain purified ionizable lipids as colorless oils. No apparent changes in the ionizable lipid integrity were observed during these procedures.

### N-oxidation of ionizable lipids

Lipids with tertiary amines (0.15 mmol) were dissolved in ethanol (1.5 mL), followed by a dropwise addition of hydrogen peroxide (0.6 mmol). The mixture was then stirred at 50 °C for 24 h. The reaction was monitored using $^1$H-NMR and HPLC/MS. The solvent was removed *in vacuo* to obtain amine oxide as a colorless oil.

### In vitro mRNA delivery

HEK-293T cells were seeded at $2 \times 10^4$ cells/well in 96-well plates. After 24 h of incubation, the cells were transfected with FLuc mRNA carrying LNPs (100 ng mRNA per well) for 24 h. Lipofectamine MessengerMAX (Thermo Fisher Scientific) was used as a positive control. To measure the transfection efficiency, cells were treated with 300 μg/mL beetle luciferin potassium salt in D-PBS (-), and relative luciferase bioluminescence was evaluated using an EnSight Multimode Plate Reader (PerkinElmer, MA, USA).

### Animals

Six–nine-week-old C57BL/6 mice and Balb/c mice were obtained from Japan SLC (Shizuoka, Japan) or Charles River Laboratories (Kanagawa, Japan). The mice were housed in groups (five mice per cage) in a specific animal facility at Hokkaido University with a 12-h day/night cycle. The mice had free access to water and a pelleted mouse diet (5053, LabDiet, USA). All experimental procedures were approved by the Hokkaido University Animal Care Committee and followed the Guidelines for the Care and Use of Laboratory Animals (approval number: 20-0176).

### ELISpot assay

C57BL/6 mice (9 weeks old, female) were intramuscularly administered 1 μg OVA mRNA-loaded LNPs into the left thigh muscles twice at 3-week intervals. At 1-week post-boost, the animals were euthanized by cervical dislocation, and the spleens were collected and placed in RPMI 1640 medium (R5886, Sigma-Aldrich, St Louis, MO, USA). The spleen of each animal was dissociated using a gentleMACS Octo Dissociator with Heaters (Miltenyi Biotec, Bergisch Gladbach, Germany), filtered through a 40 μm cell strainer, and washed. The pellets were treated with ACK Lysing buffer (Thermo Fisher Scientific) for 2 min at 25 °C and washed. The cells were resuspended in RPMI 1640 medium supplemented with 10% FBS (10270-160, Gibco, Grand Island, NY, USA) and 1% L-Glutamine-Penicillin-Streptomycin solution (G1146, Sigma-Aldrich). OVA-specific cellular immune responses were investigated using a Mouse IFN-γ ELISPOT Kit (R&D systems, Minneapolis, MN, USA). Cells ($3.75 \times 10^4$) were incubated with 10 μM OVA class I peptide (SIINFEKL, Sigma-Aldrich) at a 5% CO$_2$ atmosphere and 37 °C for 20 h, and IFN-γ secreting cells were visualized according to the manufacturer's protocol. The spots were counted using an ImmunoSpot S6 Analyzer (Cellular Technology Limited, Cleveland, OH, USA).

### ELISA for anti-OVA IgG titer

C57BL/6 mice (9 weeks old, female) were intramuscularly administered 1 μg OVA mRNA-loaded LNPs into the left thigh muscles twice at 3-week intervals. At 3 weeks post-boost, blood was collected in MiniCollect® CAT Serum tubes (Greiner Bio-One, USA) from the heart under isoflurane anesthesia and incubated for 30 min and centrifuged at $3000 \times g$ for 10 min at 25 °C to collect serum. The Nunc immuno plate was covered with 100 μL of 100 μg/mL Ovalbumin (A2512, Sigma-Aldrich) in sodium carbonate buffer overnight at 4 °C. The plate was washed thrice with PBS containing 0.05% Tween-20 (PBS-T), blocked with 200 μL of 10 g/L Block ACE (UKB80, KAC Co., Kyoto, Japan), and incubated at 4 °C overnight. After the plate was washed thrice with PBS-T, 50 μL of diluted serum with 4 g/L Block ACE was added to the well and incubated at 25 °C for 90 min. Subsequently, the plate was washed thrice with PBS-T and incubated with 100 μL of

diluted HRP conjugated goat anti-mouse IgG-Fc fragment antibody (A90-131P, Bethyl Laboratories, Montgomery, TX, USA) in 4 g/L Block ACE at 25 °C for 60 min. The plate was washed thrice with PBS-T and incubated with 100 μL of TMB solution (Nacalai tesque, Kyoto, Japan) at 25 °C for 30 min. The reaction was stopped by adding 100 μL of 1 M sulfuric acid solution, and absorbance at 450 nm was measured using the EnSight Multimode Plate Reader (PerkinElmer).

### Quantification of luciferase bioluminescence intensity

The LNPs loaded with FLuc mRNA were intravenously injected at a dose of 0.1 mg kg$^{-1}$ into Balb/c mice (6–8 weeks old, female). At 6 h post dosing, the mice were injected with VivoGlo Luciferin In Vivo Grade (Promega, Madison, WI, USA) dissolved in PBS ($-$) via the tail vein at a concentration of 1.5 mg per mouse. Three minutes after the injection, the mice were euthanized by cervical dislocation, and the liver of each animal was collected and imaged using an in vivo imaging system 200 series (PerkinElmer), according to the manufacturer's protocol.

### Quantification of secreted hEPO protein level

The LNPs loaded with hEPO mRNA were intravenously injected at 0.25 mg kg$^{-1}$ into Balb/c mice (6–8 weeks old, female). At 6 h post dosing, blood was collected from the heart under isoflurane anesthesia, incubated for 30 min, and centrifuged at $1000 \times g$ for 15 min at 25 °C to collect serum. The mice serum was stored at $-80$ °C until analysis. Samples were measured using a Human EPO ELISA Kit (Thermo Fisher Scientific), according to the manufacturer's protocol. The hEPO protein level was calculated using a four-parameter logistic curve fit analysis.

### Statistics and reproducibility

Technically independent experiments and biologically independent animals per group were defined as the number of replicates for in vitro and in vivo experiments, respectively. The results are presented as the mean ± SD unless otherwise stated. Data were analyzed using the paired Student's t-test or one-way ANOVA with Dunnett's test or Tukey's test for multiple comparisons in GraphPad Prism 6 (GraphPad Corp, San Diego, CA, USA).

### Reporting summary

Further information on research design is available in the Nature Portfolio Reporting Summary linked to this article.

## Data availability

The source data for the graphs and charts in the figures is available as Supplementary data Files. The data that support the plots within this paper and other findings of this study are available from the corresponding author upon reasonable request.

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

## Acknowledgements
This work was funded by the Nitto Denko Corporation, and support for research expenses was obtained from the Ministry of Education, Culture, Sports, Science and Technology. We would like to thank Editage (www.editage.com) for English language editing. The authors would also like to thank Takuya Shishido and Masao Murakawa for their critical review; Chisato Noyama for helping with the physicochemical characteristic measurements; and Eri Nishiura, Hiroko Nakatsukasa, and Masashi Morishita for their help with the HPLC studies.

## Author contributions
Conceptualization and Methodology: K.H., Y.S., A.Ok. and H.H.; Investigation, K.H., Y.S., M.T., S.S., A.Ot. and Y.M. Writing – Original Draft: K.H. Writing –Review & Editing: Y.S. and H.H. Supervision: Y.S., H.E., A.Ok. and H.H.

## Competing interests
K.H., M.T., S.S., A.Ot., Y.M., H.H. and Y.S. are the authors of patent WO2022/071582 (A1) and appl. No. JP2022-61002 in relation to this publication. K.H., M.T., S.S., A.Ot., Y.M., H.E. and A.Ok. were employees of the Nitto Denko Corporation at the time of this study.
