## [Peer review file · Communications Biology]

Reviewers' comments:

Reviewer #1 (Remarks to the Author):

Summary:

Hashiba et. al developed a ionizable lipid library that were piperidine-based to allow for mRNA/LNP to be stored as liquid at 4C with no loss in activity. The limitation of having mRNA/LNP stored in liquid form is the generation of aldehydes which will form adducts between mRNA and the ionizable lipid. As an effect, mRNA would be deactivated, and subsequent expression of protein would decrease. The authors demonstrated that the heterocyclic amine structures of their ionizable lipids prevent the generation of aldehydes by avoiding oxidation of the lipids.

Overall Impression:

The authors were very thorough in their explanation on the effects of aldehyde formation on mRNA stability. Throughout the Results, the authors had rigorous quantification of impurities: aldehyde generation and adduct formation. The presented data supported the claims and offered mechanistic insights to understand loss of mRNA function as a function of storage conditions. However, several weaknesses were noted that would greatly strengthen the broad applicability of the work.

Comments:

1. The use of piperidine alternatives is impactful and intriguing. However, the authors should compare their structures and the impact of the work to similar published structures, such as those made with piperazine derivatives. Even though there is a difference in the chemical structures between piperzine and piperidine, this discussion would put the work into context of recent advances in the field.
2. The authors stated that activity of LNPs decreased over time (Fig. 2C). However, there is a modest increase in EPO content in some groups after 150 days compared to 100 days. The authors should discuss why this increase (although likely statistically insignificant) might be occurring.
3. Figure 4C is missing the guanosine structure.
4. Figure 4H compares two LNP groups (15F and 4F). However it is difficult to compare formulations with the same main chain and side chain (ex. CL15F 10- 8 vs. CL4F 10-8). It would be interesting to also show this data directly comparing the same main and side chains to understand how chain length impacts delivery for each of the ionizable lipids.
5. The authors stated that IFN-gamma secretion was significantly increased compared to ALC-0315 and SM-102. These lipids were used to compare for efficacy, but weren't compared mechanistically to understand their storage stability. The authors should use these ionizable lipids as control groups for some of the mechanistic studies looking at aldehyde generation and adduct formation to compare to the piperidine ionizable lipids.
6. Figure 4F has three datapoints, making it difficult to interpret the adduct formation rate. The adduct formation plateaus, but the rate is also important. The authors should include additional timepoints prior to 24 hours.
7. With the new ionizable lipid structures, it is important to ensure that their ionization potential is still within the desirable range for mRNA delivery. The authors should provide data on ionization of the new ionizable lipids.
8. A critical part of mRNA vaccines is the use of nucleoside modifications. In this manuscript, the authors used 5-methoxyuridine modifications. The use of modifications, and the type of modification, can impact mRNA stability and how it interacts with the lipids. The authors should investigate (or at least comment on) how mRNA modifications impact their findings in the article. Similarly, does the mRNA sequence (i.e. how many of each base pair) impact their findings?
9. There is a typo in the introduction – "LNPs and their components can cause physical and chemical

damage during storage.”

10. Figure 1D – It is not clear which ionizable lipids yield higher or lower luciferase delivery. It would be good to show or describe how ionizable lipid structure impacts the level of delivery.

11. The authors should describe how they decided which ionizable lipids to mechanistically study.

The manuscript states that they selected liver-targeted lipids, but it isn't clear why these are deemed to be “liver-targeted.” How did the authors decide which to test?

12. For EPO delivery and other mechanistic studies, the authors should compare storage conditions to lyophilized LNPs as well.

13. The authors should include simple toxicity studies in vitro or in vivo looking at the piperidine ionizable lipids.

Reviewer #2 (Remarks to the Author):

Hashiba et al. constructed piperidine-based ionizable lipids to improve the thermostability of mRNA/LNP systems. They revealed that the role of amine moieties of ionizable lipids in designing mRNA/LNP. This work has important research values, however, specific comments have been described below for considerations.

1. The low stability of mRNA/LNP led to low vaccine efficacy, and therefore the efficacy of mRNA/LNP systems should be an important evaluation for their stability. However, only Figure 1 described the in vivo efficacy of mRNA/LNP, and some ionized lipids have not been determined in vaccinated mice. For instance, CL15F 12-10 and CL15F 14-12 were detected to exhibit relatively high stability but the vaccine efficacy of these two lipids with mRNA were unknown.

2. It can be useful if authors find the pattern of the main and side chain lengths of ionizable lipids for delivering mRNA to possess excellent thermostability and efficacy of mRNA/LNP, such as what main and side chain lengths contributes to highest thermostability and efficacy. Current results seem to be hard to make a conclusion.

3. It is also hard to understand why authors selected CL4F, CL5F, CL6F and CL7F with different main and side chain lengths to compare their stability. And even authors selected different main and side chain lengths of the same ionized lipids in detecting different functions in different figures. Authors should give some explanation about their design.

4. The quality of figures should be improved. For instance, panel C is missing in Figure 4.

Reviewer #3 (Remarks to the Author):

This is a very well written manuscript that deals with a very timely and pertinent topic. I suggest to publish it after a few minor revisions:

1. Why have particularly piperidine-based IL been developed? The rationale for choosing this particular structure remains elusive

2. What is the zeta potential of the LNP? Please add to Table S2

3. Fig. 1d: How did the chemical structure of the IL correlate with the transfection efficacy? This is a very relevant point and should be discussed.

Fig. 2c: Why have only CL15F 12-10 and 14-12 been tested here? What happened to the other newly synthesized IL?

Tabl S1: relative or fold-expression will be easier to digest than absolute values

Reviewer #1 (Remarks to the Author):

Summary:

Hashiba et. al developed a ionizable lipid library that were piperidine-based to allow for mRNA/LNP to be stored as liquid at 4C with no loss in activity. The limitation of having mRNA/LNP stored in liquid form is the generation of aldehydes which will form adducts between mRNA and the ionizable lipid. As an effect, mRNA would be deactivated, and subsequent expression of protein would decrease. The authors demonstrated that the heterocyclic amine structures of their ionizable lipids prevent the generation of aldehydes by avoiding oxidation of the lipids.

Overall Impression:

The authors were very thorough in their explanation on the effects of aldehyde formation on mRNA stability. Throughout the Results, the authors had rigorous quantification of impurities: aldehyde generation and adduct formation. The presented data supported the claims and offered mechanistic insights to understand loss of mRNA function as a function of storage conditions. However, several weaknesses were noted that would greatly strengthen the broad applicability of the work.

Response:

Thank you very much for your time and efforts in reviewing our manuscript. We also appreciate your positive feedback regarding our new lipid library, which suppresses aldehyde generation and limits mRNA inactivation during storage in liquid form. Your specific and helpful comments allowed us to include additional data and improve the quality of our manuscript and figures as described in Responses_1–13.

Comments:

1. The use of piperidine alternatives is impactful and intriguing. However, the authors should compare their structures and the impact of the work to similar published structures, such as those made with piperazine derivatives. Even though there is a difference in the chemical structures between piperazine and piperidine, this discussion would put the work into context of recent advances in the field.

Response_1: Thank you for providing these insights. Since C-N bonds in ionizable lipids are cleaved by oxidation and hydrolysis, the generated aldehyde structure depends on the position of the nitrogen atom in the hetero ring of the ionizable lipid. Both N-methylpiperidine CL15F lipid and the additionally synthesized CL6F lipid have heterocyclic amines, but CL6F lipid, which has a scaffold directly bound to the nitrogen atom, generated fatty aldehydes, formed undesired adducts with mRNA, and significantly decreased mRNA efficacy during storage. Based on our observation, we can predict

that lipids with scaffolds directly bound to a nitrogen atom, including piperazine lipids (Nat Commun 13, 4766 (2022). doi: 10.1038/s41467-022-32281-5; Bioeng Transl Med 8(6) (2023). doi: 10.1002/btm2.10556; Biomaterials 301:122243 (2023). doi: 10.1016/j.biomaterials.2023.122243), have the potential to yield fatty aldehyde impurities. However, this needs to be experimentally investigated. The above content was included in the Discussion section on page 22, lines 453–462 in the revised manuscript to place the work into context of the latest advances in the field as follows:

“In this study, we synthesized a novel lipid library with *N*-methyl piperidine head groups. From studies comparing *in vivo* mRNA delivery before and after storage, we demonstrated that piperidine-based lipids enabled long-term storage of mRNA/LNPs at 4 °C. ~~Notably, the nitrogen atom in the piperidine structure should not be directly bonded with the scaffold structure.~~ To provide a better understanding of the underlying mechanism of this discovery, we performed HPLC analysis and additional lipid synthesis. **Since C-N bonds in ionizable lipids are cleaved by oxidation and hydrolysis, the generated aldehyde structure depends on the position of the nitrogen atom in the hetero ring of the ionizable lipid. Both *N*-methylpiperidine CL15F lipid and the additionally synthesized CL6F lipid have heterocyclic amines; however, CL6F lipid, which has a scaffold directly bound to the nitrogen atom, yielded fatty aldehydes, formed undesired adducts with mRNA, and significantly decreased mRNA efficacy during storage. Based on our observation, it is plausible that impurified fatty aldehyde generation occurs from lipid structures such as the recently discovered piperazine derivatized lipids (Nat Commun 13, 4766 (2022). doi: 10.1038/s41467-022-32281-5; Bioeng Transl Med 8(6) (2023). doi: 10.1002/btm2.10556; Biomaterials 301:122243 (2023). doi: 10.1016/j.biomaterials.2023.122243), but this needs further experimental verification in the future. Nevertheless, this study revealed that amine moieties of ionizable lipids play a vital role in controlling reactive aldehyde generation, mRNA–lipid adduct formation, and loss of mRNA function during the storage of mRNA/LNPs.”**

2. The authors stated that activity of LNPs decreased over time (Fig. 2C). However, there is a modest increase in EPO content in some groups after 150 days compared to 100 days. The authors should discuss why this increase (although likely statistically insignificant) might be occurring.

Response_2: As you pointed out, there is indeed an increase in EPO levels from day90 to day150. Considering the overall increase, the cause may have been an unintended technical and biological variation during ELISA in the day150 experimental set. Nevertheless, we can confirm that the change is not statistically significant and does not affect our conclusions.

3. Figure 4C is missing the guanosine structure.

Response_3: The lipid-guanosine structure of the adduct has been added in Figure 4C as follows:

Figure 4

4. Figure 4H compares two LNP groups (15F and 4F). However it is difficult to compare formulations with the same main chain and side chain (ex. CL15F 10- 8 vs. CL4F 10-8). It would be interesting to also show this data directly comparing the same main and side chains to understand how chain length impacts delivery for each of the ionizable lipids.

Response_4: We directly compared two head groups with the same main chain and side chain length as shown below. In case of fresh LNPs (Day0), we observed that total carbon number (main chain length + side chain length) is important for mRNA delivery. Of note, the optimal total carbon number for the CL4F and CL15F groups is different. On the other hand, no relationship was found between tail structure and storage stability of mRNA/LNPs at 4°C (Day60 vs Day0). It is clearer that the lipid head group and the aldehyde impurity amount (Figure 3B, 4H) may be more important than the lipid tail structure as a factor for storage stability of mRNA/LNPs. This finding is consistent with Moderna's report that adduct formation depends on lot differences of ionizable lipids, including N-oxide content and aldehyde impurities rather than ionizable lipid structure, emphasizing the importance of strategic design of head structures of ionizable lipids to limit generation of reactive fatty aldehydes even if subjected to oxidation and hydrolysis.

Comparison of gene expression levels between CL4 and CL15 ionizable lipids with identical scaffolds.

Left and center figures show hepatic Fluc activity of CL4 and CL15 on Day0, respectively. Right figure indicates the percentage of relative hepatic Fluc activity of CL4 and CL15 ionizable lipids on Day60 compared with that on Day0.

5. The authors stated that IFN-gamma secretion was significantly increased compared to ALC-0315 and SM-102. These lipids were used to compare for efficacy, but weren't compared mechanistically to understand their storage stability. The authors should use these ionizable lipids as control groups for some of the mechanistic studies looking at aldehyde generation and adduct formation to compare to the piperidine ionizable lipids.

Response_5: We included ALC-0315 as a control group in a study to mechanistically examine adduct formation as shown below (Figure 4f). Results showed that ALC-0315 exhibited a similar profile to the linear amine CL4F group. Taken together with the aldehyde generation from ALC-0315 (Figure 3b) and the loss of mRNA function during storage of mRNA/ALC-0315 LNPs (Figure 2c) described in the original manuscript, the result emphasizes the importance of cyclic amines over linear amines for enhancing storage stability. We have added the sentences and experimental data in the Results section on page 14, lines 265–268 in the revised manuscript.

Figure 4 f) Percentage of adduct formation calculated from the late-eluting peak areas relative to the total peak area.

6. Figure 4F has three datapoints, making it difficult to interpret the adduct formation rate. The adduct formation plateaus, but the rate is also important. The authors should include additional timepoints prior to 24 hours.

Response_6: Accordingly, we have included two additional time points, allowing us to interpret the rate of adduct formation as shown below (Figure 4f). Since the reaction rate between mRNA and aldehyde impurity depends on their concentrations, adduct formation was detected at 1 h and further proceeded rapidly over time in the CL4F group containing a high amount of fatty aldehyde. In contrast,

a limited amount of adduct was found in the CL15F group. It should be noted that the rate of adduct formation depends on experimental conditions, including pH, temperature, and shaking speed. We have added this information and experimental data in the Results section on page 14, lines 265–268 in the revised manuscript as follows:

“To quantify adduct formation, the late-eluting peak areas were expressed as a relative percentage of the total peak area. Figure 4f shows how adduct formation proceeds over time. Adduct formation was detected at 1 h and further proceeded rapidly over time in the linear amine lipids including the CL4F group and ALC-0315 containing a high amount of fatty aldehyde. In contrast, limited amounts of adduct were observed in the CL15F group.”

Figure 4 f) Percentage of adduct formation calculated from the late-eluting peak areas relative to the total peak area.

7. With the new ionizable lipid structures, it is important to ensure that their ionization potential is still within the desirable range for mRNA delivery. The authors should provide data on ionization of the new ionizable lipids.

Response_7: We have performed the TNS assay for all LNPs and elucidated their ionization profiles at different pH values as shown below. We confirmed that all LNPs were positively charged in a pH-dependent manner. The apparent pKa values were between 6.24–7.15, which is ideal for mRNA delivery. We have added this information and experimental data in the Results section on page 5, lines 105–107 in the revised manuscript as follows:

“While LNPs containing CL15F 6-2, which had the lowest total carbon number, aggregated during the formation process owing to poor hydrophobic interactions, all other CL15F lipids were well formulated without any issues. All LNPs were positively charged in a pH-dependent manner. The apparent pKa values were between 6.24–7.15, which is ideal for mRNA delivery (Figure S1).”

Figure S1. Ionization ability of CL15-LNPs. Percentage of ionized lipid in different pH environments determined by TNS assay.

a–c) For easier visibility, the 22 LNP ionization profiles have been divided into three graphs.

8. A critical part of mRNA vaccines is the use of nucleoside modifications. In this manuscript, the authors used 5-methoxyuridine modifications. The use of modifications, and the type of modification, can impact mRNA stability and how it interacts with the lipids. The authors should investigate (or at least comment on) how mRNA modifications impact their findings in the article. Similarly, does the mRNA sequence (i.e. how many of each base pair) impact their findings?

Response_8: Since there is minimal reaction between dsRNA and aldehydes (Biochim Biophys Acta. 29(2):410-7. (1958) doi:10.1016/0006-3002(58)90200-2; J. Mol. Biol. 1(2):111-126 (1959) doi:10.1016/S0022-2836(59)80040-1), the sequence (e.g. GC-content) and modification patterns that contribute to double-strand formation can affect adduct formation efficiency. It is also known that mRNAs inside LNPs have more restricted higher-order structure than free mRNAs (J Pharm Sci. 111(3):690-698. (2022) doi: 10.1016/j.xphs.2021.11.004; Mol Pharm. 19(7):2022-2031. (2022) doi: 10.1021/acs.molpharmaceut.2c00092.). Therefore, ionizable lipid structure, lipid composition, and LNP preparation process will change the higher-order structure of mRNA and affect adduct formation. Currently, it is difficult to accurately estimate these factors and simply discuss the reactivity of each nucleoside in LNPs, but it will be important to design formulations with these factors in mind in the

future. We have added this information in the Discussion section on page 23, lines 492–500 in the revised manuscript as follows:

“Nonetheless, further studies are required to assess the generality of this proposal. **Third, the sequence (e.g. GC-content) and modification patterns that would contribute to double-strand formation can affect adduct formation efficiency since there is a small reaction between dsRNA and aldehydes (Biochim Biophys Acta. 29(2):410-7. (1958) doi:10.1016/0006-3002(58)90200-2; J. Mol. Biol. 1(2):111-126 (1959) doi:10.1016/S0022-2836(59)80040-1). It is also known that mRNAs inside LNPs have more restricted higher-order structures than free mRNAs (J Pharm Sci. 111(3):690-698. (2022) doi: 10.1016/j.xphs.2021.11.004; Mol Pharm. 19(7):2022-2031. (2022) doi: 10.1021/acs.molpharmaceut.2c00092.). Therefore, ionizable lipid structure, lipid composition, and the LNP preparation process are likely to change the higher-order structure of mRNA and affect adduct formation. Currently, it is difficult to accurately estimate the impact of these factors and simply discuss the reactivity of each nucleoside in LNPs, but is important to design formulations with these factors in mind in the future. Fourth, the loss of mRNA function can occur not only due to the formation of mRNA–lipid adducts but also because of in-line hydrolysis (J. Am. Chem. Soc. 121, 5364–5372 (1999) doi: org/10.1021/ja990592p).**

9. There is a typo in the introduction – “LNPs and their components can cause physical and chemical damage during storage.”

Response_9: The verb has been changed from “cause” to “undergo” in the Introduction section on page 3, line 57 in the revised manuscript.

10. Figure 1D – It is not clear which ionizable lipids yield higher or lower luciferase delivery. It would be good to show or describe how ionizable lipid structure impacts the level of delivery.

Response_10: Accordingly, we have clarified which ionizable lipid yielded *in vitro* luciferase mRNA delivery (Fig. S2a) as shown below. To elucidate structure–activity relationships (SARs), the lipid structure was described by two parameters (total carbon number and symmetry of the tail). Utilizing these structural parameters, we visualized the impact of lipid structure on luciferase activity as a contour plot (Fig. S2b), showing that ionizable lipids with branching and longer tails can enhance functional delivery of mRNA, consistent with our previous study (Small Sci. 3, 2200071 (2023). doi:10.1002/smsc.202200071). We have added this information and experimental data in the Results

section on page 5, lines 112–116 in the revised manuscript as follows:

“We first evaluated the *in vitro* functional delivery of firefly luciferase (FLuc) mRNA using CL15F LNPs. Most CL15F LNPs induced a more intense bioluminescence than CL4F LNPs in HEK-293T cells (Figure 1d, S2a). In addition, some CL15F LNPs exhibited an efficacy comparable to that of Lipofectamine MessengerMAX, a reagent specifically optimized for mRNA transfection *in vitro*. To elucidate structure–activity relationships (SARs), each lipid structure was described by two parameters (total carbon number and symmetry of the tail). The impact of lipid structure on luciferase activity was visualized as a contour plot (Fig. S2b), confirming that ionizable lipids with branching and longer tails can enhance functional delivery of mRNA.”

Figure S2 Impact of scaffold structure of CL15 on *in vitro* functional delivery of luciferase mRNA.

a, b) *In vitro* functional delivery of luciferase mRNA summarized as a table (a) and contour plot (b). The level is expressed as relative activity (%) against MessengerMAX used as a positive control.

The lipid tail structure is described by two parameters (total carbon number and symmetry) based on the main chain and side chain lengths of ionizable lipid tail. The total carbon number for the branched tail is calculated as (main chain length) + (side chain length). The symmetry of the main chain and side chain is calculated as the ratio of (side chain length) to (main chain length – 2). For example, CL15F 9–7, CL15F 12–4, and CL15F 16–0 have the same total carbon number (16) in each tail. In contrast, they are fully symmetrical (1.0), moderately symmetrical (0.4), and linear (0.0), respectively (y-axis).

11. The authors should describe how they decided which ionizable lipids to mechanistically study. The manuscript states that they selected liver-targeted lipids, but it isn’t clear why these are

deemed to be “liver-targeted.” How did the authors decide which to test?

Response_11: We utilized the hEPO reporter system as a simple assay for evaluating the storage stability of mRNA/LNPs *in vivo* according to published literature (Biomaterials. 286:121570 (2022). Doi: 10.1016/j.biomaterials.2022.121570). To compare the *in vivo* efficacy before and after mRNA/LNP storage, liver-targeted LNPs are desirable as they can achieve hEPO levels sufficient for quantitative evaluation. Therefore, we used CL15F 12-10 and CL15F 14-12 among lipids in the CL15F lipid library, which can efficiently deliver mRNA into the liver (see below). For the same reason, we utilized CL4F 10-6 and CL4F 11-9, which can efficiently deliver mRNA into the liver, and the CL15 counterparts with consistent scaffold structures, CL15F 10-6 and CL15F 11-9, for comparison. To clarify our experimental design, we have added this information in the Results section on page 9, lines 149–152 in the revised manuscript as follows:

“The storage stability of mRNA/LNP formulations remains a significant challenge, particularly at refrigeration temperatures. We utilized the hEPO reporter system as a simple assay for evaluating the storage stability of mRNA/LNPs. (Biomaterials. 286:121570 (2022). Doi: 10.1016/j.biomaterials.2022.121570.) To compare the *in vivo* efficacy before and after mRNA/LNP storage, liver-targeted LNPs are desirable as they can induce hEPO levels sufficient for quantitative evaluation. Therefore, We examined the stability of CL15F LNPs loaded with human erythropoietin (hEPO) mRNA over time at different temperatures (Figure 2a). CL15F 12-10 and CL15F 14-12 were selected as the liver targeting lipids and compared them with CL4F 8-6, CL4F 10-4, SM-102 and ALC-0315. To assess the *in vivo* efficacy of hEPO mRNA/LNPs, serum hEPO levels were quantified using ELISA following LNP administration at 0.25 mg/kg hEPO. hEPO secretion was confirmed in mice treated with fresh mRNA/LNPs, and this efficacy was comparable to that reported earlier (Proc Natl Acad Sci U S A. 118(52):e2109256118 (2021). Doi:10.1073/pnas.2109256118; ACS Nano. 17(3):2588-2601 (2023). Doi:10.1021/acsnano.2c10501) (Figure S1).”

Ex vivo bioluminescence in the liver was measured 6 h after the administration of 0.1 mg/kg Fluc mRNA/LNPs. N= 3, biologically independent mice per group.

12. For EPO delivery and other mechanistic studies, the authors should compare storage conditions to lyophilized LNPs as well.

Response_12: Self-amplifying RNA vaccines as lyophilized LNP formulations have been approved by Japanese authorities (medRxiv 2023.07.13.23292597 doi: 10.1101/2023.07.13.23292597). Thus, we consider lyophilized LNPs important in enhancing storage stability, and we agree with your suggestion. However, the focus of this study was on the structures of new ionizable lipids for the inhibition of aldehyde generation and the related mechanism, not on the superiority of the liquid formulations over their lyophilized counterparts. Additionally, our findings are important not only for liquid formulations but also for lyophilized formulations since adduct formation starts immediately after formulation. We are currently engaged in new research on the application of our new lipids with less aldehyde impurity generation for lyophilized LNPs. For instance, we have obtained new knowledge on the relationship between lipid structure, size changes, and mRNA leakage after reconstitution of lyophilized LNPs. The comparison of lyophilized LNPs with liquid formulations will be considered for submission in the future.

13. The authors should include simple toxicity studies *in vitro* or *in vivo* looking at the piperidine ionizable lipids.

Response_13: We have performed single dose toxicity studies *in vivo*. No *in vivo* toxicity was observed, including body weight change, and abnormalities in serum biochemistry parameters and histological scores as shown below. These results have led to the inclusion of a new Figure S4. We have also added this information in the Results section on page 9, lines 165–167 in the revised manuscript as follows:

“CL15F LNPs maintained their *in vivo* activity after 5 months of storage even at 4 °C as a liquid formulation, whereas the activity of other LNPs decreased over time, with a half-life of approximately 2 months under consistent storage conditions (Figure 2c). No *in vivo* toxicity was observed, including body weight change, and abnormal serum biochemistry parameters and histological scores at the consistent mRNA dose (Fig. S4).”

Figure S4. Single dose toxicity studies *in vivo*

a–c) Body weight measurement (a), hematological test (b), hematoxylin-eosin staining of

liver and spleen (c), and histopathological analysis (d) was performed 24 h after i.v. administration of CL15F 14-12 LNPs (0.25 mg/kg Fluc mRNA per mouse, n= 4 biologically independent balb/c mice per group). A) Body weight change was not significant based on paired Student's t-test. B) Serum chemistry parameters were measured at Oriental yeast Co., Ltd (Shiga, Japan). Serum chemistry parameters were not significantly changed based on unpaired Student's t-test. TP: total protein, ALB: albumin, BUN: blood urea nitrogen, CRE: creatinine, IP: inorganic phosphates, AST: aspartate transaminase, ALT: alanine transaminase, LDH: lactose dehydrogenase, AMY: amylase, γ -GT: gamma-glutamyl transpeptidase, T-CHO: total cholesterol, TG: triglyceride, HDL-C: high density lipoprotein cholesterol, T-BIL: total bilirubin, GLU: glucose. C) Liver and spleen were fixed in Mildform 10N and 3 μ m slices were stained with hematoxylin-eosin. Scale bars represent 100 μ m and 500 μ m for liver and spleen, respectively. D) Histopathological analysis of liver and spleen was performed at the Sapporo General Pathology Laboratory Co., Ltd. (Hokkaido, Japan).

Reviewer #2 (Remarks to the Author):

Hashiba et al. constructed piperidine-based ionizable lipids to improve the thermostability of mRNA/LNP systems. They revealed that the role of amine moieties of ionizable lipids in designing mRNA/LNP. This work has important research values, however, specific comments have been described below for considerations.

Response: Thank you very much for your time and effort in reviewing our manuscript. We express our gratitude to this reviewer for finding our work important and providing professional comments. To improve the quality of our manuscript, we have included additional data as described in Responses_1–4.

1. The low stability of mRNA/LNP led to low vaccine efficacy, and therefore the efficacy of mRNA/LNP systems should be an important evaluation for their stability. However, only Figure 1 described the *in vivo* efficacy of mRNA/LNP, and some ionized lipids have not been determined in vaccinated mice. For instance, CL15F 12-10 and CL15F 14-12 were detected to exhibit relatively high stability but the vaccine efficacy of these two lipids with mRNA were unknown.

Response_1: It would be interesting but challenging to correlate mRNA/LNP storage stability with vaccine efficacy, since the relationships between antigen expression level and immune response is unclear. (Mol Ther Nucleic Acids. 15:1-11 (2019). Doi: 10.1016/j.omtn.2019.01.013.) Lipid structure, mRNA modifications, and dsRNA impurity levels also affect vaccine efficacy. (Nat Biotechnol. 37(10):1174-1185 (2019). Doi: 10.1038/s41587-019-0247-3.; Proc Natl Acad Sci U S A. 120(29):e2214320120 (2023). Doi: 10.1073/pnas.2214320120.). Therefore, we confirmed vaccine efficacy using fresh mRNA/LNPs immediately after preparation, while storage stability of mRNA/LNPs was investigated by a simple evaluation system (Fluc or hEPO) according to published literature (Biomaterials. 286:121570 (2022). Doi: 10.1016/j.biomaterials.2022.121570.; Nat Commun. 12(1):6777 (2021). Doi: 10.1038/s41467-021-26926-0.)

It is difficult to relate vaccine efficacy to storage stability, but we completely agree with your suggestion that the therapeutic effect should be demonstrated for CL15F 12-10 and CL15F 14-12 LNPs, which have higher stability at the protein expression level. In fact, further investigation has identified CL15F 12-10 and CL15F 14-12 LNPs as more suitable for application in liver genome editing than in vaccines. We are in the process of exploring the application of CL15F LNPs and will consider submitting this as future work.

2. It can be useful if authors find the pattern of the main and side chain lengths of ionizable lipids for

delivering mRNA to possess excellent thermostability and efficacy of mRNA/LNP, such as what main and side chain lengths contributes to highest thermostability and efficacy. Current results seem to be hard to make a conclusion.

Response_2: The lipid tail structure can be described by two parameters (total carbon number and symmetry) based on main chain and side chain lengths of ionizable lipid tail. Utilizing these structural parameters, we visualized the impact of lipid structure on luciferase activity as a contour plot (Fig. S2b), confirming that ionizable lipids with branching and longer tails can enhance functional delivery of mRNA, consistent with our previous study (Small Sci. 3, 2200071 (2023). doi:10.1002/smsc.202200071).

Unlike the above structure–activity relationship, it seems challenging to understand structure–thermostability relationship. As illustrated in our study, the lipid head structure is associated with aldehyde generation and important for the thermostability of mRNAs/LNPs. In contrast, the lipid tail structure may alter the higher-order structure of the mRNA inside the LNPs and thus have an impact on adduct formation. Specifically, it is known that mRNAs inside LNPs have more restricted higher-order structure than free mRNAs (J Pharm Sci. 111(3):690-698. (2022) doi: 10.1016/j.xphs.2021.11.004; Mol Pharm. 19(7):2022-2031. (2022) doi: 10.1021/acs.molpharmaceut.2c00092.). It is also noted that there is minimal reaction between dsRNA and aldehydes (Biochim Biophys Acta. 29(2):410-7. (1958) doi:10.1016/0006-3002(58)90200-2; J. Mol. Biol. 1(2):111-126 (1959) doi:10.1016/S0022-2836(59)80040-1). Taken together, some impact of the lipid tail on the higher-order structure of mRNA inside LNPs is postulated. Further advances in this field are needed to investigate the structure–thermostability relationship.

We have added this information in the Results section on page 5, lines 112–116 in the revised manuscript as follows:

“We first evaluated the *in vitro* functional delivery of firefly luciferase (fLuc) mRNA using CL15F LNPs. Most CL15F LNPs induced a more intense bioluminescence than CL4F LNPs in HEK-293T cells (Figure 1d, S2a). In addition, some CL15F LNPs exhibited an efficacy comparable to that of Lipofectamine MessengerMAX, a reagent specifically optimized for mRNA transfection *in vitro*. To elucidate structure–activity relationships (SARs), each lipid structure was described by two parameters (total carbon number and tail symmetry). The impact of lipid structure on luciferase activity was visualized as a contour plot (Fig. S2b), confirming that ionizable lipids with branching and longer tails can enhance functional delivery of mRNA.”

In addition, we have added more information in the Discussion section on page 23, lines 492–500 in the revised manuscript as follows:

“Nonetheless, further studies are required to assess the generality of this proposal. Third, the sequence (e.g. GC-content) and modification patterns that would contribute to double-strand formation can affect adduct formation efficiency since there is a small reaction between dsRNA and aldehydes (Biochim Biophys Acta. 29(2):410-7. (1958) doi:10.1016/0006-3002(58)90200-2; J. Mol. Biol. 1(2):111-126 (1959) doi:10.1016/S0022-2836(59)80040-1). It is also known that mRNAs inside LNPs have more restricted higher-order structures than free mRNAs (J Pharm Sci. 111(3):690-698. (2022) doi: 10.1016/j.xphs.2021.11.004; Mol Pharm. 19(7):2022-2031. (2022) doi: 10.1021/acs.molpharmaceut.2c00092.). Therefore, ionizable lipid structure, lipid composition, and the LNP preparation process are likely to change the higher-order structure of mRNA and affect adduct formation. Currently, it is difficult to accurately estimate the impact of these factors and simply discuss the reactivity of each nucleoside in LNPs, but is important to design formulations with these factors in mind in the future. Fourth, the loss of mRNA function can occur not only due to the formation of mRNA–lipid adducts but also because of in-line hydrolysis (J. Am. Chem. Soc. 121, 5364–5372 (1999) doi: org/10.1021/ja990592p).

Figure S2. Impact of scaffold structure of CL15 on *in vitro* functional delivery of luciferase mRNA and size change during long term storage.

a, b) *In vitro* functional delivery of luciferase mRNA summarized as a table (a) and contour plot (b). The level is expressed as relative activity (%) against MessengerMAX used as a positive control.

The lipid tail structure is described by two parameters (total carbon number and symmetry) based on the main chain and side chain lengths of ionizable lipid tail. The total carbon number for the branched tail is calculated as (main chain length) + (side chain length). The symmetry of the main chain and side chain is calculated as the ratio of (side chain length)

to (main chain length – 2). For example, CL15F 9–7, CL15F 12–4, and CL15F 16–0 have the same total carbon number (16) in each tail. In contrast, they are fully symmetrical (1.0), moderately symmetrical (0.4), and linear (0.0), respectively (y-axis).

3. It is also hard to understand why authors selected CL4F, CL5F, CL6F and CL7F with different main and side chain lengths to compare their stability. And even authors selected different main and side chain lengths of the same ionized lipids in detecting different functions in different figures. Authors should give some explanation about their design.

Response_3: mRNA/LNPs lose efficacy due to physical damage of LNPs and chemical modification of mRNA. To focus on the chemical modification of mRNA such as lipid–mRNA adducts, we mainly used longer and branched tail lipids, which have the most potential to physically stabilize LNPs (Small Sci. 3, 2200071 (2023). doi:10.1002/smsc.202200071). Therefore, we synthesized CL4F 14-12, CL15F 14-12, CL6F 14-12, CL16F 14-12, and CL17F 14-12 (Figure 6). All LNPs were physically stable, allowing us to focus on the impact of the head structure on aldehyde generation, adduct formation, and mRNA inactivation (Table S3).

In other experiments, we mainly selected longer and branched tail lipids such as CL15F 11-9, CL15F 10-6, CL15F 12-10, and CL15F 14-12 for the above reason. In some cases, other lipids were utilized to increase the number of examples (n) and enhance generality. In Fig. 3c and Fig. 4a–d, the lipid with high relative aldehyde impurities was utilized to facilitate the identification of the aldehyde structure and its adduct structure.

To clarify our experimental design, we have added this information in the Results section on page 20, lines 393–395 and page 11, lines 202–203 in the revised manuscript as follows:

“To further investigate the role of lipid structure in aldehyde generation, adduct formation, and storage stability, we synthesized additional lipids with three different cyclic amine moieties as the head structure (Figure 6a). We selected a long and symmetrical tail, 14-12, with the most potential to physically stabilize LNPs (Small Sci. 3, 2200071 (2023). doi:10.1002/smsc.202200071) to rule out the possibility that LNPs may physicochemically deteriorate. From the above possible pathways (Figure 5i), CL16F 14-12 and CL17F 14-12 do not generate reactive fatty aldehydes, even if oxidation and hydrolysis occur, because they can form 5- or 6-membered cyclic imines via intramolecular aldehyde–amine reactions (Figure 6b). In contrast, CL6F 14-12, although heterocyclic, has the potential to deactivate mRNA because it has a nitrogen atom directly bonded to the scaffold, which can generate fatty byproducts containing only aldehyde groups as well as linear amine lipids.”

“Carbonyl compounds reacted with DNPH to form stable hydrazones, which were detected using a UV detector; this enabled the detection of small amounts of aldehyde impurities in the samples. CL4F 10-8 and CL4F 11-5, which exhibited higher values in the NBD-H assay, were selected as samples. The absorbance at 360 nm in the UV chromatogram revealed the presence of DNPH-derivatized carbonyl compounds in each sample (Figure 3c).”

4. The quality of figures should be improved. For instance, panel C is missing in Figure 4.

Response_4: The lipid-guanosine structure of the adduct has been added in Figure 4 panel C as follows:

Figure 4

Reviewer #3 (Remarks to the Author):

This is a very well written manuscript that deals with a very timely and pertinent topic. I suggest to publish it after a few minor revisions:

Response: Thank you very much for your time and efforts in reviewing our manuscript. We also appreciate your positive feedback on our work about piperidine-based ionizable lipids to overcome thermostability challenges in mRNA/LNPs. Your constructive comments allowed us to improve the quality of our manuscript and figures as described in Responses_1–5.

1. Why have particularly piperidine-based IL been developed? The rationale for choosing this particular structure remains elusive

Response_1: In earlier work on siRNA delivery, *N*-methyl piperidine was identified as one of the promising head groups (J Control Release. 295:140-152 (2019). Doi: 10.1016/j.jconrel.2019.01.001). It was also expected that heterocyclic amine-containing lipids would induce a strong immune response in mRNA/LNPs (Nat Biotechnol. 37(10):1174-1185 (2019). Doi: 10.1038/s41587-019-0247-3.). Therefore, to enhance functional mRNA delivery, a variety of branched structures were introduced into our new lipid library. We have added this information in the Results section on page 5, lines 88–92 in the revised manuscript as follows:

“A library of 23 novel ionizable lipids with *N*-methyl piperidine head groups was designed for mRNA delivery (Figure 1a). In earlier work on siRNA delivery, *N*-methyl piperidine was identified as one of the promising head groups (J Control Release. 295:140-152 (2019). Doi: 10.1016/j.jconrel.2019.01.001). In addition, heterocyclic amine-containing lipids were reported to elicit a strong immune response in mRNA/LNPs (Nat Biotechnol. 37(10):1174-1185 (2019). Doi: 10.1038/s41587-019-0247-3). Therefore, to enhance functional mRNA delivery, a variety of branched structures were introduced into our new lipid library. ~~These lipids include branched fatty acids as the hydrophobic tails for facilitating endosomal escape and improving the physical stability of LNPs⁴⁹.~~

2. What is the zeta potential of the LNP? Please add to Table S2

Response_2: We have included zeta potential data on mRNA/LNPs in the revised supporting information. Owing to a response to another comment, Table S2 is now Table S1.

3. Fig. 1d: How did the chemical structure of the IL correlate with the transfection efficacy? This is a very relevant point and should be discussed.

Response_3: To understand structure–activity relationships (SARs), the lipid structure was described by two parameters (total carbon number and symmetry of the tail). Utilizing these structural parameters, we visualized the impact of lipid structure on luciferase activity as a contour plot (Fig. S2b), confirming that ionizable lipids with branching and longer tails can enhance functional delivery of mRNA, consistent with our previous study (Small Sci. 3, 2200071 (2023). doi:10.1002/smsc.202200071). We have added this information in the Results section on page 5, lines 112–116 in the revised manuscript as follows:

“We first evaluated the *in vitro* functional delivery of firefly luciferase (FLuc) mRNA using CL15F LNPs. Most CL15F LNPs induced a more intense bioluminescence than CL4F LNPs in HEK-293T cells (Figure 1d, S2a). In addition, some CL15F LNPs exhibited an efficacy comparable to that of Lipofectamine MessengerMAX, a reagent specifically optimized for mRNA transfection *in vitro*. To elucidate structure–activity relationships (SARs), each lipid structure was described by two parameters (total carbon number and symmetry of the tail). The impact of lipid structure on luciferase activity was visualized as a contour plot (Fig. S2b), confirming that ionizable lipids with branching and longer tails can enhance functional delivery of mRNA.”

Figure S2. Impact of scaffold structure of CL15 on *in vitro* functional delivery of luciferase mRNA.

a, b) *In vitro* functional delivery of luciferase mRNA summarized as a table (a) and contour plot (b). The level is expressed as relative activity (%) against MessengerMAX used as a positive control.

The lipid tail structure is described by two parameters (total carbon number and symmetry)

based on the main chain and side chain lengths of ionizable lipid tail. The total carbon number for the branched tail is calculated as (main chain length) + (side chain length). The symmetry of the main chain and side chain is calculated as the ratio of (side chain length) to (main chain length - 2). For example, CL15F 9-7, CL15F 12-4, and CL15F 16-0 have the same total carbon number (16) in each tail. In contrast, they are fully symmetrical (1.0), moderately symmetrical (0.4), and linear (0.0), respectively (y-axis).

4. Fig. 2c: Why have only CL15F 12-10 and 14-12 been tested here? What happened to the other newly synthesized IL?

Response_4: We utilized the hEPO reporter system as a simple assay for evaluating the storage stability of mRNA/LNPs *in vivo* according to published literature (Biomaterials. 286:121570 (2022). doi: 10.1016/j.biomaterials.2022.121570.). To compare the *in vivo* efficacy before and after mRNA/LNP storage, liver-targeted LNPs are desirable as they can achieve hEPO levels sufficient for quantitative evaluation. Therefore, we used CL15F 12-10 and CL15F 14-12 among lipids in the CL15F lipid library, which can efficiently deliver mRNA to the liver (see). We further demonstrated that the five lipids with CL15F head structure enhanced storage stability, even in the luciferase assay (Fig. 4H). To clarify our experimental design, we have added this information in the Results section on page 9, lines 149-152 in the revised manuscript as follows:

“The storage stability of mRNA/LNP formulations remains a significant challenge, particularly at refrigeration temperatures. We utilized the hEPO reporter system as a simple assay for evaluating the storage stability of mRNA/LNPs. (Biomaterials. 286:121570 (2022). doi: 10.1016/j.biomaterials.2022.121570.) To compare the *in vivo* efficacy before and after mRNA/LNP storage, liver-targeted LNPs are desirable as they can induce hEPO levels sufficient for quantitative evaluation. Therefore, We examined the stability of CL15F LNPs loaded with human erythropoietin (hEPO) mRNA over time at different temperatures (Figure 2a). CL15F 12-10 and CL15F 14-12 were selected as the liver-targeting lipids and compared them with CL4F 8-6, CL4F 10-4, SM-102, and ALC-0315.”

Ex vivo bioluminescence in the liver was measured 6 h after the administration of 0.1 mg/kg Fluc mRNA/LNPs. n= 3, biologically independent mice per group.

5. Tabl S1: relative or fold-expression will be easier to digest than absolute values

Response_5: Accordingly, relative instead of absolute values have been included in the table. Due to a response to another comment, Table S1 has now become panel A of Fig S2 as follows:

Figure S2 Impact of scaffold structure of CL15 on *in vitro* functional delivery of luciferase mRNA.

a, b) *In vitro* functional delivery of luciferase mRNA summarized as a table (a) and contour plot (b). The level is expressed as relative activity (%) against MessengerMAX used as a positive control.

The lipid tail structure is described by two parameters (total carbon number and symmetry) based on main chain and side chain lengths of ionizable lipid tail. The total carbon number for the branched tail is calculated as (main chain length) + (side chain length). The symmetry of the main chain and side chain is calculated as the ratio of (side chain length)

to (main chain length - 2). For example, CL15F 9-7, CL15F 12-4, and CL15F 16-0 have the same total carbon number (16) in each tail. In contrast, they are fully symmetrical (1.0), moderately symmetrical (0.4), and linear (0.0), respectively (y-axis).

REVIEWERS' COMMENTS:

Reviewer #1 (Remarks to the Author):

The revisions were thoroughly considered and the required experiments were performed. The data supports their claims.

Reviewer #2 (Remarks to the Author):

Authors have addressed my previous comments. Therefore I recommend acceptance of the revised manuscript.

Reviewer #3 (Remarks to the Author):

All comments have been sufficiently addressed.